# A Definition of Continual Reinforcement Learning

**David Abel**
dmabel@google.com
Google DeepMind

**André Barreto**
andrebarreto@google.com
Google DeepMind

**Benjamin Van Roy**
benvanroy@google.com
Google DeepMind

**Doina Precup**
doinap@google.com
Google DeepMind

**Hado van Hasselt**
hado@google.com
Google DeepMind

**Satinder Singh**
baveja@google.com
Google DeepMind

## Abstract

In a standard view of the reinforcement learning problem, an agent's goal is to efficiently identify a policy that maximizes long-term reward. However, this perspective is based on a restricted view of learning as *finding a solution*, rather than treating learning as *endless adaptation*. In contrast, continual reinforcement learning refers to the setting in which the best agents never stop learning. Despite the importance of continual reinforcement learning, the community lacks a simple definition of the problem that highlights its commitments and makes its primary concepts precise and clear. To this end, this paper is dedicated to carefully defining the continual reinforcement learning problem. We formalize the notion of agents that "never stop learning" through a new mathematical language for analyzing and cataloging agents. Using this new language, we define a continual learning agent as one that can be understood as carrying out an implicit search process indefinitely, and continual reinforcement learning as the setting in which the best agents are all continual learning agents. We provide two motivating examples, illustrating that traditional views of multi-task reinforcement learning and continual supervised learning are special cases of our definition. Collectively, these definitions and perspectives formalize many intuitive concepts at the heart of learning, and open new research pathways surrounding continual learning agents.

## 1   Introduction

In *The Challenge of Reinforcement Learning*, Sutton states: "Part of the appeal of reinforcement learning is that it is in a sense the whole AI problem in a microcosm" [56]. Indeed, the problem facing an agent that learns to make better decisions from experience is at the heart of the study of Artificial Intelligence (AI). Yet, when we study the reinforcement learning (RL) problem, it is typical to restrict our focus in a number of ways. For instance, we often suppose that a complete description of the state of the environment is available to the agent, or that the interaction stream is subdivided into episodes. Beyond these standard restrictions, however, there is another significant assumption that constrains the usual framing of RL: We tend to concentrate on agents that learn to solve problems, rather than agents that learn forever. For example, consider an agent learning to play Go: Once the agent has discovered how to master the game, the task is complete, and the agent's learning can stop. This view of learning is often embedded in the standard formulation of RL, in which an agent interacts with a Markovian environment with the goal of efficiently identifying an optimal policy, at which point learning can cease.

But what if this is not the best way to model the RL problem? That is, instead of viewing learning as *finding a solution*, we can instead think of it as *endless adaptation*. This suggests study of the *continual* reinforcement learning (CRL) problem [47, 48, 25, 27], as first explored in the thesis by

Ring [46], with close ties to supervised never-ending [10, 39, 43] and continual learning [47, 48, 26, 54, 41, 42, 49, 22, 30, 45, 4].

Despite the prominence of CRL, the community lacks a clean, general definition of this problem. It is critical to develop such a definition to promote research on CRL from a clear conceptual foundation, and to guide us in understanding and designing continual learning agents. To these ends, this paper is dedicated to carefully defining the CRL problem. Our definition is summarized as follows:

### The CRL Problem (Informal)
*An RL problem is an instance of CRL if the best agents never stop learning.*

The core of our definition is framed around two new insights that formalize the notion of "agents that never stop learning": (i) we can understand *every agent* as implicitly searching over a set of history-based policies (Theorem 3.1), and (ii) *every agent* will either continue this search forever, or eventually stop (Remark 3.2). We make these two insights rigorous through a pair of logical operators on agents that we call *generates* and *reaches* that provide a new mathematical language for characterizing agents. Using these tools, we then define CRL as any RL problem in which all of the best agents never stop their implicit search. We provide two motivating examples of CRL, illustrating that traditional multi-task RL and continual supervised learning are special cases of our definition. We further identify necessary properties of CRL (Theorem 4.1) and the new operators (Theorem 4.2, Theorem 4.3). Collectively, these definitions and insights formalize many intuitive concepts at the heart of continual learning, and open new research pathways surrounding continual learning agents.

## 2 Preliminaries

We first introduce key concepts and notation. Our conventions are inspired by Ring [46], the recent work by Dong et al. [16] and Lu et al. [32], as well as the literature on *general RL* by Hutter [23, 24], Lattimore [28], Leike [29], Cohen et al. [12], and Majeed [36].

**Notation.** We let capital calligraphic letters denote sets ($\mathcal{X}$), lower case letters denote constants and functions ($x$), italic capital letters denote random variables ($X$), and blackboard capitals denote the natural and real numbers ($\mathbb{N}, \mathbb{R}, \mathbb{N}_0 = \mathbb{N} \cup \{0\}$). Additionally, we let $\Delta(\mathcal{X})$ denote the probability simplex over the set $\mathcal{X}$. That is, the function $p : \mathcal{X} \times \mathcal{Y} \to \Delta(\mathcal{Z})$ expresses a probability mass function $p(\cdot \mid x, y)$, over $\mathcal{Z}$, for each $x \in \mathcal{X}$ and $y \in \mathcal{Y}$. Lastly, we use $\neg$ to denote logical negation, and we use $\forall_{x \in \mathcal{X}}$ and $\exists_{x \in \mathcal{X}}$ to express the universal and existential quantifiers over a set $\mathcal{X}$.

### 2.1 Agents and Environments

We begin by defining environments, agents, and related artifacts.

**Definition 2.1.** *An agent-environment **interface** is a pair $(\mathcal{A}, O)$ of countable sets $\mathcal{A}$ and $O$ where $|\mathcal{A}| \geq 2$ and $|O| \geq 1$.*

We refer to elements of $\mathcal{A}$ as *actions*, denoted $a$, and elements of $O$ as *observations*, denoted $o$. Histories define the possible interactions between an agent and an environment that share an interface.

**Definition 2.2.** *The **histories** with respect to interface $(\mathcal{A}, O)$ are the set of sequences of action-observation pairs,*

$$\mathcal{H} = \bigcup_{t=0}^{\infty} (\mathcal{A} \times O)^t . \tag{2.1}$$

We refer to an individual element of $\mathcal{H}$ as a *history*, denoted $h$, and we let $hh'$ express the history resulting from the concatenation of any two histories $h, h' \in \mathcal{H}$. Furthermore, the set of histories of length $t \in \mathbb{N}_0$ is defined as $\mathcal{H}_t = (\mathcal{A} \times O)^t$, and we use $h_t \in \mathcal{H}_t$ to refer to a history containing $t$ action-observation pairs, $h_t = a_0 o_1 \ldots a_{t-1} o_t$, with $h_0 = \emptyset$ the empty history. An environment is then a function from the set of all environments, $\mathcal{E}$, that produces observations given a history.

**Definition 2.3.** *An **environment** with respect to interface $(\mathcal{A}, O)$ is a function $e : \mathcal{H} \times \mathcal{A} \to \Delta(O)$.*

This model of environments is general in that it can capture Markovian environments such as Markov decision processes (MDPs, Puterman, 2014) and partially observable MDPs (Cassandra et al., 1994), as well as both episodic and non-episodic settings. We next define an agent as follows.

**Definition 2.4.** *An **agent** with respect to interface $(\mathcal{A}, \mathcal{O})$ is a function $\lambda : \mathcal{H} \to \Delta(\mathcal{A})$.*

We let $\mathbb{A}$ denote the set of all agents, and let $\Lambda$ denote any non-empty subset of $\mathbb{A}$. This treatment of an agent captures the mathematical way experience gives rise to behavior, as in "agent functions" from work by Russell and Subramanian [50]. This is in contrast to a mechanistic account of agency as proposed by Dong et al. [16] and Sutton [58]. Further, note that Definition 2.4 is precisely a history-based policy; we embrace the view that there is no real distinction between an agent and a policy, and will refer to all such functions as "agents" unless otherwise indicated.

## 2.2 Realizable Histories

We will be especially interested in the histories that occur with non-zero probability as a result of the interaction between a particular agent and environment.

**Definition 2.5.** *The **realizable histories** of a given agent-environment pair, $(\lambda, e)$, define the set of histories of any length that can occur with non-zero probability from the interaction of $\lambda$ and $e$,*

$$\mathcal{H}^{\lambda,e} = \bar{\mathcal{H}} = \bigcup_{t=0}^{\infty} \left\{ h_t \in \mathcal{H}_t : \prod_{k=0}^{t-1} e(o_{k+1} \mid h_k, a_k) \lambda(a_k \mid h_k) > 0 \right\}. \tag{2.2}$$

Given a realizable history $h$, we will refer to the realizable history *suffixes*, $h'$, which, when concatenated with $h$, produce a realizable history $hh' \in \bar{\mathcal{H}}$.

**Definition 2.6.** *The **realizable history suffixes** of a given $(\lambda, e)$ pair, relative to a history prefix $h \in \mathcal{H}^{\lambda,e}$, define the set of histories that, when concatenated with prefix $h$, remain realizable,*

$$\mathcal{H}_h^{\lambda,e} = \bar{\mathcal{H}}_h = \{ h' \in \mathcal{H} : hh' \in \mathcal{H}^{\lambda,e} \}. \tag{2.3}$$

We abbreviate $\mathcal{H}^{\lambda,e}$ to $\bar{\mathcal{H}}$, and $\mathcal{H}_h^{\lambda,e}$ to $\bar{\mathcal{H}}_h$, where $\lambda$ and $e$ are obscured for brevity.

## 2.3 Reward, Performance, and the RL Problem

Supported by the arguments of Bowling et al. [7], we assume that all of the relevant goals or purposes of an agent are captured by a deterministic reward function (in line with the *reward hypothesis* [57]).

**Definition 2.7.** *We call $r : \mathcal{A} \times \mathcal{O} \to \mathbb{R}$ a **reward function**.*

We remain agnostic to how the reward function is implemented; it could be a function inside of the agent, or the reward function's output could be a special scalar in each observation. Such commitments do not impact our framing. When we refer to an environment we will implicitly mean that a reward function has been selected as well. We remain agnostic to how reward is aggregated to determine performance, and instead adopt the function $v$ defined as follows.

**Definition 2.8.** *The **performance**, $v : \mathcal{H} \times \mathbb{A} \times \mathcal{E} \to [v_{\min}, v_{\max}]$ is a bounded function for fixed constants $v_{\min}, v_{\max} \in \mathbb{R}$.*

The function $v(\lambda, e \mid h)$ expresses some statistic of the received future random rewards produced by the interaction between $\lambda$ and $e$ following history $h$, where we use $v(\lambda, e)$ as shorthand for $v(\lambda, e \mid h_0)$. While we accommodate any $v$ that satisfies the above definition, it may be useful to think of specific choices of $v(\lambda, e \mid h_t)$, such as the average reward,

$$\liminf_{k \to \infty} \frac{1}{k} \mathbb{E}_{\lambda,e}[R_t + \ldots + R_{t+k} \mid H_t = h_t], \tag{2.4}$$

where $\mathbb{E}_{\lambda,e}[\, \cdots \mid H_t = h_t]$ denotes expectation over the stochastic process induced by $\lambda$ and $e$ following history $h_t$. Or, we might consider performance based on the expected discounted reward, $v(\lambda, e \mid h_t) = \mathbb{E}_{\lambda,e}[R_t + \gamma R_{t+1} + \ldots \mid H_t = h_t]$, where $\gamma \in [0, 1)$ is a discount factor.

The above components give rise to a simple definition of the RL problem.

**Definition 2.9.** *An instance of the **RL problem** is defined by a tuple $(e, v, \Lambda)$ as follows*

$$\Lambda^* = \arg\max_{\lambda \in \Lambda} v(\lambda, e). \tag{2.5}$$

This captures the RL problem facing an *agent designer* that would like to identify an optimal agent $(\lambda^* \in \Lambda^*)$ with respect to the performance $(v)$, among the available agents $(\Lambda)$, in a particular environment $(e)$. We note that a simple extension of this definition of the RL problem might instead consider a set of environments (or similar alternatives).

# 3    Agent Operators: Generates and Reaches

We next introduce two new insights about agents, and the logical operators that formalize them:

1. Theorem 3.1: *Every agent* can be understood as searching over another set of agents.
2. Remark 3.2: *Every agent* will either continue their search forever, or eventually stop.

We make these insights precise by introducing a pair of logical operators on agents: (1) a set of agents *generates* (Definition 3.4) another set of agents, and (2) a given agent *reaches* (Definition 3.5) an agent set. Together, these operators enable us to define *learning* as the implicit search process captured by the first insight, and *continual learning* as the process of continuing this search indefinitely.

## 3.1    Operator 1: An Agent Basis Generates an Agent Set.

The first operator is based on two complementary intuitions.

From the first perspective, an agent can be understood as *searching* over a space of representable action-selection strategies. For instance, in an MDP, agents can be interpreted as searching over the space of policies (that is, the space of stochastic mappings from the MDP's state to action). It turns out this insight can be extended to any agent and any environment.

The second complementary intuition notes that, as agent designers, we often first identify the space of representable action-selection strategies of interest. Then, it is natural to design agents that search through this space. For instance, in designing an agent to interact with an MDP, we might be interested in policies representable by a neural network of a certain size and architecture. When we design agents, we then consider all agents (choices of loss function, optimizer, memory, and so on) that search through the space of assignments of weights to this particular neural network using standard methods like gradient descent. We codify these intuitions in the following definitions.

**Definition 3.1.** *An **agent basis** (or simply, a basis), $\Lambda_B \subset \mathbb{A}$, is any non-empty subset of $\mathbb{A}$.*

Notice that an agent basis is a choice of agent set, $\Lambda$. We explicitly call out a basis with distinct notation ($\Lambda_B$) as it serves an important role in the discussion that follows. For example, we next introduce *learning rules* as functions that switch between elements of an agent basis for each history.

**Definition 3.2.** *A **learning rule** over an agent basis $\Lambda_B$ is a function, $\sigma : \mathcal{H} \rightarrow \Lambda_B$, that selects a base agent for each history.*

We let $\bar{\Sigma}$ denote the set of all learning rules over $\Lambda_B$, and let $\Sigma$ denote any non-empty subset of $\bar{\Sigma}$. A learning rule is a mechanism for switching between the available base agents following each new experience. Notice that learning rules are deterministic; while a simple extension captures the stochastic case, we will see by Theorem 3.1 that the above is sufficiently general in a certain sense. We use $\sigma(h)(h)$ to refer to the action distribution selected by the agent $\lambda = \sigma(h)$ at any history $h$.

**Definition 3.3.** *Let $\Sigma$ be a set of learning rules over some basis $\Lambda_B$, and let $e$ be an environment. We say that a set $\Lambda$ is $\Sigma$-**generated** by $\Lambda_B$ in $e$, denoted $\Lambda_B \Vdash_\Sigma \Lambda$, if and only if*

$$\forall_{\lambda \in \Lambda} \exists_{\sigma \in \Sigma} \forall_{h \in \bar{\mathcal{H}}} \quad \lambda(h) = \sigma(h)(h). \tag{3.1}$$

Thus, any choice of $\Sigma$ together with a basis $\Lambda_B$ induces a family of agent sets whose elements can be understood as switching between the basis according to the rules prescribed by $\Sigma$. We then say that a basis *generates* an agent set in an environment if there exists a set of learning rules that switches between the basis elements to produce the agent set.

**Definition 3.4.** *We say a basis $\Lambda_B$ **generates** $\Lambda$ in $e$, denoted $\Lambda_B \Vdash \Lambda$, if and only if*

$$\Lambda_B \Vdash_{\bar{\Sigma}} \Lambda. \tag{3.2}$$

Intuitively, an agent basis $\Lambda_B$ generates another agent set $\Lambda$ just when the agents in $\Lambda$ can be understood as switching between the base agents. It is in this sense that we can understand agents as searching through a basis—an agent is just a particular sequence of history-conditioned switches over a basis. For instance, let us return to the example of a neural network: The agent basis might represent a specific multilayer perceptron, where each element of this basis is an assignment to the network's weights. The learning rules are different mechanisms that choose the next set of weights in

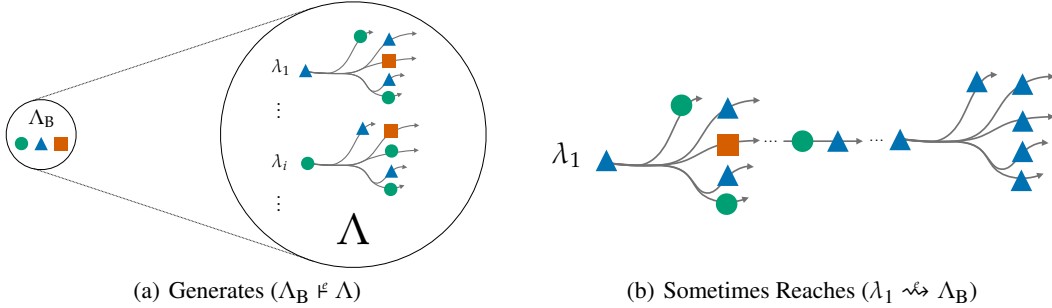

(a) Generates ($\Lambda_B \not\Vdash \Lambda$)   (b) Sometimes Reaches ($\lambda_1 \rightsquigarrow \Lambda_B$)

Figure 1: A visual of the generates (left) and sometimes reaches (right) operators. **(a) Generates:** An agent basis, $\Lambda_B$, comprised of three base agents depicted by the triangle, circle, and square, generates a set $\Lambda$ containing agents that can each be understood as switching between the base agents in the realizable histories of environment $e$. **(b) Sometimes Reaches:** On the right, we visualize $\lambda_1 \in \Lambda$ generated by $\Lambda_B$ (from the figure on the left) to illustrate the concept of *sometimes reaches*. That is, the agent's choice of action distribution at each history can be understood as switching between the three basis elements, and there is at least one history for which the agent stops switching—here, we show the agent settling on the choice of the blue triangle and never switching again.

response to experience (such as gradient descent). Together, the agent basis and the learning rules *generate* the set of agents that search over choices of weights in reaction to experience. We present a cartoon visual of the generates operator in Figure 1(a).

Now, using the generates operator, we revisit and formalize the central insight of this section: Every agent can be understood as implicitly searching over an agent basis. We take this implicit search process to be the behavioral signature of learning.

**Theorem 3.1.** *For any agent-environment pair ($\lambda$, $e$), there exists infinitely many choices of a basis, $\Lambda_B$, such that both (1) $\lambda \notin \Lambda_B$, and (2) $\Lambda_B \not\Vdash \{\lambda\}$.*

Due to space constraints, all proofs are deferred to Appendix B.

We require that $\lambda \notin \Lambda_B$ to ensure that the relevant bases are non-trivial generators of $\{\lambda\}$. This theorem tells us that no matter the choice of agent or environment, we can view the agent as a series of history-conditioned switches between basis elements. In this sense, we can understand the agent *as if*[1] it were carrying out a search over the elements of some $\Lambda_B$. We emphasize that there are infinitely many choices of such a basis to illustrate that there are many plausible interpretations of an agent's behavior—we return to this point throughout the paper.

### 3.2 Operator 2: An Agent Reaches a Basis.

Our second operator reflects properties of an agent's limiting behavior in relation to a basis. Given an agent and a basis that the agent searches through, what happens to the agent's search process in the limit: does the agent keep switching between elements of the basis, or does it eventually stop? For example, in an MDP, many agents of interest eventually stop their search on a choice of a fixed policy. We formally define this notion in terms of an agent *reaching* a basis according to two modalities: an agent (i) *sometimes* or (ii) *never* reaches a basis.

**Definition 3.5.** *We say agent $\lambda \in \wedge$ **sometimes reaches** $\Lambda_B$ in e, denoted $\lambda \rightsquigarrow \Lambda_B$, if and only if*

$$\exists_{h \in \bar{\mathcal{H}}} \exists_{\lambda_B \in \Lambda_B} \forall_{h' \in \bar{\mathcal{H}}_h} \quad \lambda(hh') = \lambda_B(hh'). \tag{3.3}$$

That is, for at least one realizable history, there is some base agent ($\lambda_B$) that produces the same action distribution as $\lambda$ forever after. This indicates that the agent can be understood as if it has stopped its search over the basis. We present a visual of sometimes reaches in Figure 1(b). By contrast, we say an agent *never* reaches a basis just when it never becomes equivalent to a base agent.

**Definition 3.6.** *We say agent $\lambda \in \wedge$ **never reaches** $\Lambda_B$ in e, denoted $\lambda \not\rightsquigarrow \Lambda_B$, iff $\neg(\lambda \rightsquigarrow \Lambda_B)$.*

---

[1]We use *as if* in the sense of the positive economists, such as Friedman [19].

The reaches operators formalize the intuition that, since every agent can be interpreted as if it were searching over a basis, *every agent* will either (1) sometimes, or (2) never stop this search. Since (1) and (2) are simply negations of each other, we can now plainly state this fact as follows.

**Remark 3.2.** *For any agent-environment pair $(\lambda, e)$ and any choice of basis $\Lambda_B$ such that $\Lambda_B \vDash \{\lambda\}$, exactly one of the following two properties must be satisfied:*

$$(1)\ \lambda \rightsquigarrow \Lambda_B, \qquad (2)\ \lambda \not\rightsquigarrow \Lambda_B. \tag{3.4}$$

Thus, by Theorem 3.1, every agent can be thought of as implicitly searching over an agent basis, and by Remark 3.2, every agent will either (1) sometimes, or (2) never stop this search. We take this implicit search process to be the behavioral signature of learning, and will later exploit this perspective to define a continual learning agent as one that continues its search forever (Definition 4.1). Our analysis in Section 4.4 further elucidates basic properties of both the generates and reaches operators, and Figure 1 presents a cartoon visualizing the intuition behind each of the operators. We summarize all definitions and notation in a table in Appendix A.

**Considerations on the Operators.** Naturally, we can design many variations of both $\rightsquigarrow$ and $\vDash$. For instance, we might be interested in a variant of reaches in which an agent becomes $\epsilon$-close to any of the basis elements, rather than require exact behavioral equivalence. Concretely, we highlight four axes of variation that can modify the definitions of the operators. We state these varieties for reaches, but similar modifications can be made to the generates operator, too:

1. *Realizability.* An agent reaches a basis (i) in all histories (and thus, all environments), or (ii) in the histories realizable by a given $(\lambda, e)$ pair.

2. *History Length.* An agent reaches a basis over (i) infinite or, (ii) finite length histories.

3. *Probability.* An agent reaches a basis (i) with probability one, or (ii) with high probability.

4. *Equality or Approximation.* An agent reaches a basis by becoming (i) equivalent to a base agent, or (ii) sufficiently similar to a base agent.

Rather than define all of these variations precisely for both operators (though we do explore some in Appendix C), we acknowledge their existence, and simply note that the formal definitions of these variants follow naturally.

## 4   Continual Reinforcement Learning

We now provide a precise definition of CRL. The definition formalizes the intuition that CRL captures settings in which the best agents *do not converge*—they continue their implicit search over an agent basis indefinitely.

### 4.1   Definition: Continual RL

We first define continual learning agents using the generates and never reaches operators as follows.

**Definition 4.1.** *An agent $\lambda$ is a **continual learning agent** in $e$ relative to $\Lambda_B$ if and only if the basis generates the agent ($\Lambda_B \vDash \{\lambda\}$) and the agent never reaches the basis ($\lambda \not\rightsquigarrow \Lambda_B$).*

This means that an agent is a continual learner in an environment relative to $\Lambda_B$ if the agent's search over $\Lambda_B$ continues forever. Notice that an agent might be considered a continual learner with respect to one basis but not another; we explore this fact more in Section 4.4.

Then, using these tools, we formally define CRL as follows.

> **Definition 4.2.** *Consider an RL problem $(e, v, \Lambda)$. Let $\Lambda_B \subset \Lambda$ be a basis such that $\Lambda_B \vDash \Lambda$, and let $\Lambda^* = \arg\max_{\lambda \in \Lambda} v(\lambda, e)$. We say $(e, v, \Lambda, \Lambda_B)$ defines a **CRL problem** if $\forall_{\lambda^* \in \Lambda^*} \lambda^* \not\rightsquigarrow \Lambda_B$.*

Said differently, an RL problem is an instance of CRL just when all of the best agents are continual learning agents relative to basis $\Lambda_B$. This problem encourages a significant departure from how we tend to think about designing agents: Given a basis, rather than try to build agents that can solve problems by identifying a fixed high-quality element of the basis, we would like to design agents that continue to update their behavior indefinitely in light of their experience.

### 4.2 CRL Examples

We next detail two examples of CRL to provide further intuition.

**Q-Learning in Switching MDPs.** First we consider a simple instance of CRL based on the standard multi-task view of MDPs. In this setting, the agent repeatedly samples an MDP to interact with from a fixed but unknown distribution [64, 9, 2, 25, 20]. In particular, we make use of the switching MDP environment from Luketina et al. [33]. The environment $e$ consists of a collection of $n$ underlying MDPs, $m_1, \ldots, m_n$, with a shared action space and environment-state space. We refer to this environment-state space using observations, $o \in O$. The environment has a fixed constant positive probability of 0.001 to switch the underlying MDP, which yields different transition and reward functions until the next switch. The agent only observes each environment state $o \in O$, which does not reveal the identity of the active MDP. The rewards of each underlying MDP are structured so that each MDP has a unique optimal policy. We assume $v$ is defined as the average reward, and the basis is the set of $\epsilon$-greedy policies over all $Q(o, a)$ functions, for fixed $\epsilon = 0.15$. Consequently, the set of agents we generate, $\Lambda_{\mathrm{B}} \nvDash \Lambda$, consists of all agents that switch between these $\epsilon$-greedy policies.

Now, the components $(e, v, \Lambda, \Lambda_{\mathrm{B}})$ have been defined, we can see that this is indeed an instance of CRL: None of the base agents can be optimal, as the moment that the environment switches its underlying MDP, we know that any previously optimal policy will no longer be optimal in the next MDP following the switch. Therefore, any agent that *converges* (in that it reaches the basis $\Lambda_{\mathrm{B}}$) cannot be optimal either for the same reason. We conclude that all optimal agents in $\Lambda$ are continual learning agents relative to the basis $\Lambda_{\mathrm{B}}$.

We present a visual of this domain in Figure 2(a), and conduct a simple experiment contrasting the performance of $\epsilon$-greedy *continual* Q-learning (blue) that uses a constant step-size parameter of $\alpha = 0.1$, with a *convergent* Q-learning (green) that anneals its step size parameter over time to zero. Both use $\epsilon = 0.15$, and we set the number of underlying MDPs to $n = 10$. We present the average reward with 95% confidence intervals, averaged over 250 runs, in Figure 2(b). Since both variants of Q-learning can be viewed as searching over $\Lambda_{\mathrm{B}}$, the annealing variant that stops its search will under-perform compared to the continual approach. These results support the unsurprising conclusion that it is better to continue searching over the basis rather than converge in this setting.

**Continual Supervised Learning.** Second, we illustrate the power of our CRL definition to capture continual supervised learning. We adopt the problem setting studied by Mai et al. [35]. Let $\mathcal{X}$ denote a set of objects to be labeled, each belonging to one of $k \in \mathbb{N}$ classes. The observation space $O$ consists of pairs, $o_t = (x_t, y_t)$, where $x_t \in \mathcal{X}$ and $y_t \in \mathcal{Y}$. Here, each $x_t$ is an input object to be classified and $y_t$ is the label for the previous input $x_{t-1}$. Thus, $O = \mathcal{X} \times \mathcal{Y}$. We assume by convention that the initial

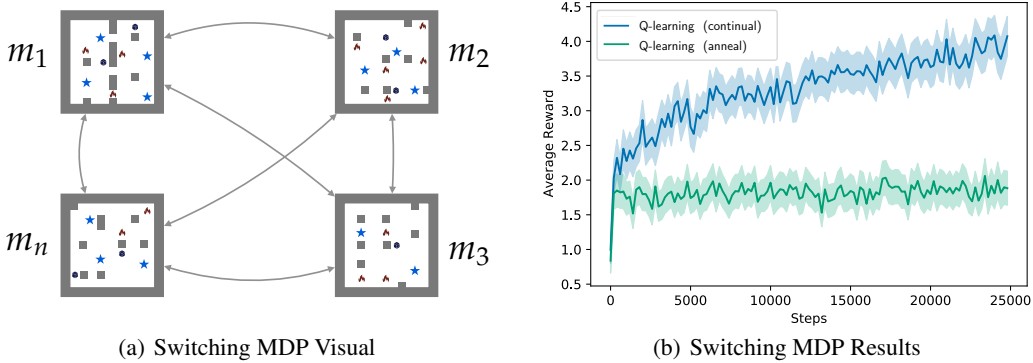

(a) Switching MDP Visual          (b) Switching MDP Results

Figure 2: A visual of a grid world instance of the switching MDPs problem (left) [33], and results from an experiment contrasting continual learning and convergent Q-learning (right). The environment pictured contains $n$ distinct MDPs. Each underlying MDP shares the same state space and action space, but varies in transition and reward functions, as indicated by the changing walls and rewarding locations (stars, circles, and fire). The results pictured on the right contrast continual Q-learning (with $\alpha = 0.1$) with traditional Q-learning that anneals its step-size parameter to zero over time.

label $y_0$ is irrelevant and can be ignored. The agent will observe a sequence of object-label pairs, $(x_0, y_0), (x_1, y_1), \ldots$, and the action space is a choice of label, $\mathcal{A} = \{a_1, \ldots, a_k\}$ where $|\mathcal{Y}| = k$. The reward for each history $h_t$ is $+1$ if the agent's most recently predicted label is correct for the previous input, and $-1$ otherwise:

$$r(a_{t-1}o_t) = r(a_{t-1}y_t) = \begin{cases} +1 & a_{t-1} = y_t, \\ -1 & a_{t-1} = \text{otherwise}. \end{cases} \qquad (4.1)$$

Concretely, the continual learning setting studied by Mai et al. [35] supposes the learner will receive samples from a sequence of probability distributions, $d_0, d_1, \ldots$, each supported over $\mathcal{X} \times \mathcal{Y}$. The $(x, y) \in \mathcal{X} \times \mathcal{Y}$ pairs experienced by the learner are determined by the sequence of distributions. We capture this distributional shift in an environment $e$ that shifts its probability distribution over $O$ depending on the history to match the sequence, $d_0, d_1, \ldots$.

Now, is this an instance of CRL? To answer this question precisely, we need to select a $(\Lambda, \Lambda_B)$ pair. We adopt the basis $\Lambda_B = \{\lambda_B : x \mapsto y_i, \forall_{y_i \in \mathcal{Y}}\}$ that contains each classifier that maps each object to each possible label. By the universal set of learning rules $\Sigma$, this basis generates the set of all agents that search over classifiers. Now, our definition says the above is an instance of CRL just when every optimal agent never stops switching between classifiers, rather than stop their search on a fixed classifier. Consequently, if there is an optimal classifier in $\Lambda_B$, then this will not be an instance of CRL. If, however, the environment imposes enough distributional shift (changing labels, adding mass to new elements, and so on), then the *only* optimal agents will be those that always switch among the base classifiers, in which case the setting is an instance of CRL.

## 4.3 Relationship to Other Views on Continual Learning

The spirit of continual learning has been an important part of machine learning research for decades, often appearing under the name of "lifelong learning" [63, 62, 53, 55, 51, 3, 4], "never-ending learning" [39, 43] with close ties to transfer-learning [61, 60], meta-learning [52, 17], as well as online learning and non-stationarity [5, 40, 13, 6, 31]. In a similar vein, the phrase "continuing tasks" is used in the classic RL textbook [59] to refer explicitly to cases when the interaction between agent and environment is not subdivided into episodes. Continual reinforcement learning was first posed in the thesis by Ring [46]. In later work [47, 48], Ring proposes a formal definition of the continual reinforcement learning problem—The emphasis of Ring's proposal is on the *generality* of the environment: rather than assume that agents of interest will interact with an MDP, Ring suggests studying the unconstrained case in which an agent must maximize performance while only receiving a stream of observations as input. The environment or reward function, in this sense, may change over time or may be arbitrarily complex. This proposal is similar in spirit to *general RL*, studied by Hutter [24], Lattimore [28], Leike [29], and others [12, 37, 36] in which an agent interacts with an unconstrained environment. General RL inspires many aspects of our conception of CRL; for instance, our emphasis on history-dependence rather than environment-state comes directly from general RL. More recently, Khetarpal et al. [25] provide a comprehensive survey of the continual reinforcement learning literature. We encourage readers to explore this survey for a detailed history of the subject.[2] In the survey, Khetarpal et al. propose a definition of the CRL problem that emphasizes the non-stationarity of the underlying process. In particular, in Khetarpal et al.'s definition, an agent interacts with a POMDP in which each of the individual components of the POMDP—such as the state space or reward function—are allowed to vary with time. We note that, as the environment model we study (Definition 2.3) is a function of history, it can capture time-indexed non-stationarity. In this sense, the same generality proposed by Khetarpal et al. and Ring is embraced and retained by our definition, but we add further precision to what is meant by *continual learning* by centering around a mathematical definition of continual learning agents (Definition 4.1).

## 4.4 Properties of CRL

Our formalism is intended to be a jumping off point for new lines of thinking around agents and continual learning. We defer much of our analysis and proofs to the appendix, and here focus on highlighting necessary properties of CRL.

---

[2]For other surveys, see the recent survey on continual robot learning by Lesort et al. [30], a survey on continual learning with neural networks by Parisi et al. [42], a survey on transfer learning in RL by Taylor and Stone [60], and a survey on continual image classification by Mai et al. [35].

**Theorem 4.1.** *Every instance of CRL $(e, v, \Lambda, \Lambda_\mathrm{B})$ necessarily satisfies the following properties:*

1. *If $\Lambda \neq \Lambda_\mathrm{B} \cup \Lambda^*$, then there exists a $\Lambda_\mathrm{B}'$ such that (1) $\Lambda_\mathrm{B}' \not\models \Lambda$, and (2) $(e, v, \Lambda, \Lambda_\mathrm{B}')$ is not an instance of CRL.*

2. *No element of $\Lambda_\mathrm{B}$ is optimal: $\Lambda_\mathrm{B} \cap \Lambda^* = \emptyset$.*

3. *If $|\Lambda|$ is finite, there exists an agent set, $\Lambda^\circ$, such that $|\Lambda^\circ| < |\Lambda|$ and $\Lambda^\circ \not\models \Lambda$.*

4. *If $|\Lambda|$ is infinite, there exists an agent set, $\Lambda^\circ$, such that $\Lambda^\circ \subset \Lambda$ and $\Lambda^\circ \not\models \Lambda$.*

This theorem tells us several things. The first point of the theorem has peculiar implications. We see that as we change a single element (the basis $\Lambda_\mathrm{B}$) of the tuple $(e, v, \Lambda_\mathrm{B}, \Lambda)$, the resulting problem can change from CRL to not CRL. By similar reasoning, an agent that is said to be a *continual learning agent* according to Definition 4.1 may not be a continual learner with respect to some other basis. We discuss this point further in the next paragraph. Point (2.) notes that no optimal strategy exists within the basis—instead, to be optimal, an agent must switch between basis elements indefinitely. As discussed previously, this fact encourages a departure in how we think about the RL problem: rather than focus on agents that can identify a single, fixed solution to a problem, CRL instead emphasizes designing agents that are effective at updating their behavior indefinitely. Points (3.) and (4.) show that $\Lambda$ cannot be *minimal*. That is, there are necessarily some redundancies in the design space of the agents in CRL—this is expected, since we are always focusing on agents that search over the same agent basis. Lastly, it is worth calling attention to the fact that in the definition of CRL, we assume $\Lambda_\mathrm{B} \subset \Lambda$—this suggests that in CRL, the agent basis is necessarily limited in some way. Consequently, the design space of agents $\Lambda$ are *also* limited in terms of what agents they can represent at any particular point in time. This limitation may come about due to a computational or memory budget, or by making use of a constrained set of learning rules. This suggests a deep connection between *bounded* agents and the nature of continual learning, as explored further by Kumar et al. [27]. While these four points give an initial character of the CRL problem, we note that further exploration of the properties of CRL is an important direction for future work.

**Canonical Agent Bases.** It is worth pausing and reflecting on the concept of an agent basis. As presented, the basis is an arbitrary choice of a set of agents—consequently, point (1.) of Theorem 4.1 may stand out as peculiar. From this perspective, it is reasonable to ask if the fact that our definition of CRL is basis-dependant renders it vacuous. We argue that this is not the case for two reasons. First, we conjecture that *any* definition of continual learning that involves concepts like "learning" and "convergence" will have to sit on top of some reference object whose choice is arbitrary. Second, and more important, even though the mathematical construction allows for an easy change of basis, in practice the choice of basis is constrained by considerations like the availability of computational resources. It is often the case that the domain or problem of interest provides obvious choices of bases, or imposes constraints that force us as designers to restrict attention to a space of plausible bases or learning rules. For example, as discussed earlier, a choice of neural network architecture might comprise a basis—any assignment of weights is an element of the basis, and the learning rule $\sigma$ is a mechanism for updating the active element of the basis (the parameters) in light of experience. In this case, the number of parameters of the network is constrained by what we can actually build, and the learning rule needs to be suitably efficient and well-behaved. We might again think of the learning rule $\sigma$ as gradient descent, rather than a rule that can search through the basis in an unconstrained way. In this sense, the basis is not *arbitrary*. We as designers choose a class of functions to act as the relevant representations of behavior, often limited by resource constraints on memory or compute. Then, we use specific learning rules that have been carefully designed to react to experience in a desirable way—for instance, stochastic gradient descent updates the current choice of basis in the direction that would most improve performance. For these reasons, the choice of basis is not arbitrary, but instead reflects the ingredients involved in the design of agents as well as the constraints necessarily imposed by the environment.

## 4.5 Properties of Generates and Reaches

Lastly, we summarize some of the basic properties of generates and reaches. Further analysis of generates, reaches, and their variations is provided in Appendix C.

**Theorem 4.2.** *The following properties hold of the generates operator:*

1. *Generates is transitive: For any triple $(\Lambda^1, \Lambda^2, \Lambda^3)$ and $e \in \mathcal{E}$, if $\Lambda^1 \Vdash \Lambda^2$ and $\Lambda^2 \Vdash \Lambda^3$, then $\Lambda^1 \Vdash \Lambda^3$.*

2. *Generates is not commutative: there exists a pair $(\Lambda^1, \Lambda^2)$ and $e \in \mathcal{E}$ such that $\Lambda^1 \Vdash \Lambda^2$, but $\neg(\Lambda^2 \Vdash \Lambda^1)$.*

3. *For all $\Lambda$ and pair of agent bases $(\Lambda_B^1, \Lambda_B^2)$ such that $\Lambda_B^1 \subseteq \Lambda_B^2$, if $\Lambda_B^1 \Vdash \Lambda$, then $\Lambda_B^2 \Vdash \Lambda$.*

4. *For all $\Lambda$ and $e \in \mathcal{E}$, $\mathbb{\Lambda} \Vdash \Lambda$.*

5. *The decision problem, **Given** $(e, \Lambda_B, \Lambda)$, **output** True iff $\Lambda_B \Vdash \Lambda$, is undecidable.*

The fact that generates is transitive suggests that the basic tools of an agent set—paired with a set of learning rules—might be likened to an algebraic structure. We can draw a symmetry between an agent basis and the basis of a vector space: A vector space is comprised of all linear combinations of the basis, whereas $\Lambda$ is comprised of all valid switches (according to the learning rules) between the base agents. However, the fact that generates is not commutative (by point 2.) raises a natural question: are there choices of learning rules under which generates is commutative? We suggest that a useful direction for future work can further explore an algebraic perspective on agents.

We find many similar properties hold of reaches.

**Theorem 4.3.** *The following properties hold of the reaches operator:*

1. *$\rightsquigarrow$ and $\not\rightsquigarrow$ are not transitive.*

2. *"Sometimes reaches" is not commutative: there exists a pair $(\Lambda^1, \Lambda^2)$ and $e \in \mathcal{E}$ such that $\forall_{\lambda^1 \in \Lambda^1} \ \lambda^1 \rightsquigarrow \Lambda^2$, but $\exists_{\lambda^2 \in \Lambda^2} \ \lambda^2 \not\rightsquigarrow \Lambda^1$.*

3. *For all pairs $(\Lambda, e)$, if $\lambda \in \Lambda$, then $\lambda \rightsquigarrow \Lambda$.*

4. *Every agent satisfies $\lambda \rightsquigarrow \mathbb{\Lambda}$ in every environment.*

5. *The decision problem, **Given** $(e, \lambda, \Lambda)$, **output** True iff $\lambda \rightsquigarrow \Lambda$, is undecidable.*

Many of these properties resemble those in Theorem 4.2. For instance, point (5.) shows that deciding whether a given agent sometimes reaches a basis in an environment is undecidable. We anticipate that the majority of decision problems related to determining properties of arbitrary agent sets interacting with unconstrained environments will be undecidable, though it is still worth making these arguments carefully. Moreover, there may be interesting special cases in which these decision problems are decidable (and perhaps, efficiently so). We suggest that identifying these special cases and fleshing out their corresponding efficient algorithms is an interesting direction for future work.

## 5  Discussion

In this paper, we carefully develop a simple mathematical definition of the continual RL problem. We take this problem to be of central importance to AI as a field, and hope that these tools and perspectives can serve as an opportunity to think about CRL and its related artifacts more carefully. Our proposal is framed around two new insights about agents: (i) every agent can be understood as though it were searching over an agent basis (Theorem 3.1), and (ii) every agent, in the limit, will either sometimes or never stop this search (Remark 3.2). These two insights are formalized through the generates and reaches operators, which provide a rich toolkit for understanding agents in a new way—for example, we find straightforward definitions of a continual learning agent (Definition 4.1) and learning rules (Definition 3.2). We anticipate that further study of these operators and different families of learning rules can directly inform the design of new learning algorithms; for instance, we might characterize the family of *continual* learning rules that are guaranteed to yield continual learning agents, and use this to guide the design of principled continual learning agents (in the spirit of continual backprop by Dohare et al. [14]). In future work, we intend to further explore connections between our formalism of continual learning and some of the phenomena at the heart of recent empirical continual learning studies, such as plasticity loss [34, 1, 15], in-context learning [8], and catastrophic forgetting [38, 18, 21, 26]. More generally, we hope that our definitions, analysis, and perspectives can help the community to think about continual reinforcement learning in a new light.

## Acknowledgements

The authors are grateful to Michael Bowling, Clare Lyle, Razvan Pascanu, and Georgios Piliouras for comments on a draft of the paper, as well as the anonymous NeurIPS reviewers that provided valuable feedback on the paper. The authors would further like to thank all of the 2023 Barbados RL Workshop participants and Elliot Catt, Will Dabney, Sebastian Flennerhag, András György, Steven Hansen, Anna Harutyunyan, Mark Ho, Joe Marino, Joseph Modayil, Rémi Munos, Evgenii Nikishin, Brendan O'Donoghue, Matt Overlan, Mark Rowland, Tom Schaul, Yannick Shroecker, Rich Sutton, Yunhao Tang, Shantanu Thakoor, and Zheng Wen for inspirational conversations.

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

# A Notation

We first provide a table summarizing all relevant notation.

| Notation | Meaning | Definition |
|---|---|---|
| $\mathcal{A}$ | Actions | |
| $O$ | Observations | |
| | | |
| $\mathcal{H}_t$ | Length $t$ histories | $\mathcal{H}_t = (\mathcal{A} \times O)^t$ |
| $\mathcal{H}$ | All histories | $\mathcal{H} = \bigcup_{t=0}^{\infty} \mathcal{H}_t$ |
| $h$ | A history | $h \in \mathcal{H}$ |
| $hh'$ | History concatenation | |
| $h_t$ | Length $t$ history | $h_t \in \mathcal{H}_t$ |
| $\bar{\mathcal{H}} = \mathcal{H}^{\lambda,e}$ | Realizable histories | $\bar{\mathcal{H}} = \bigcup_{t=0}^{\infty} \left\{ h_t \in \mathcal{H}_t : \prod_{k=0}^{t-1} e(o_k \mid h_k, a_k) \lambda(a_k \mid h_k) > 0 \right\}$ |
| $\bar{\mathcal{H}}_h = \mathcal{H}_h^{\lambda,e}$ | Realizable history suffixes | $\bar{\mathcal{H}}_h = \{ h' \in \mathcal{H} : hh' \in \mathcal{H}^{\lambda,e} \}$ |
| | | |
| $e$ | Environment | $e : \mathcal{H} \times \mathcal{A} \to \Delta(\mathcal{A})$ |
| $\mathcal{E}$ | Set of all environments | |
| | | |
| $\lambda$ | Agent | $\lambda : \mathcal{H} \to \Delta(\mathcal{A})$ |
| $\mathbb{A}$ | Set of all agents | |
| $\Lambda$ | Set of agents | $\Lambda \subseteq \mathbb{A}$ |
| $\Lambda_\mathrm{B}$ | Agent basis | $\Lambda_\mathrm{B} \subset \mathbb{A}$ |
| | | |
| $r$ | Reward function | $r : \mathcal{A} \times O \to \mathbb{R}$ |
| $v$ | Performance | $v : \mathcal{H} \times \mathbb{A} \times \mathcal{E} \to [\mathrm{v_{min}}, \mathrm{v_{max}}]$ |
| | | |
| $\sigma$ | Learning rule | $\sigma : \mathcal{H} \to \Lambda_\mathrm{B}$ |
| $\mathbb{\Sigma}$ | Set of all learning rules | |
| $\Sigma$ | Set of learning rules | $\Sigma \subseteq \mathbb{\Sigma}$ |
| | | |
| $\Lambda_\mathrm{B} \vDash_\Sigma^e \Lambda$ | $\Sigma$-generates | $\forall_\Lambda \exists_{\sigma \in \Sigma} \forall_{h \in \bar{\mathcal{H}}} \; \lambda(h) = \sigma(h)(h)$ |
| $\Lambda_\mathrm{B} \vDash^e \Lambda$ | Generates | $\exists_{\Sigma \subseteq \mathbb{\Sigma}} \Lambda_\mathrm{B} \vDash_\Sigma^e \Lambda$ |
| $\Lambda_\mathrm{B} \models_\Sigma \Lambda$ | Universally $\Sigma$-generates | $\forall_\Lambda \exists_{\sigma \in \Sigma} \forall_{h \in \mathcal{H}} \; \lambda(h) = \sigma(h)(h)$ |
| $\Lambda_\mathrm{B} \models \Lambda$ | Universally generates | $\exists_{\Sigma \subseteq \mathbb{\Sigma}} \Lambda_\mathrm{B} \models_\Sigma \Lambda$ |
| | | |
| $\lambda \rightsquigarrow \Lambda_\mathrm{B}$ | Sometimes reaches | $\exists_{h \in \bar{\mathcal{H}}} \exists_{\lambda_\mathrm{B} \in \Lambda_\mathrm{B}} \forall_{h' \in \bar{\mathcal{H}}_h} \; \lambda(hh') = \lambda_\mathrm{B}(hh')$ |
| $\lambda \not\rightsquigarrow \Lambda_\mathrm{B}$ | Never reaches | $\neg(\lambda \rightsquigarrow \Lambda_\mathrm{B})$ |
| $\lambda \;\square\!\!\rightsquigarrow \Lambda_\mathrm{B}$ | Always reaches | $\forall_{h \in \bar{\mathcal{H}}} \exists_{t \in \mathbb{N}_0} \forall_{h^\circ \in \bar{\mathcal{H}}_h^{t:\infty}} \exists_{\lambda_\mathrm{B} \in \Lambda_\mathrm{B}} \forall_{h' \in \bar{\mathcal{H}}_h} \; \lambda(hh^\circ h') = \lambda_\mathrm{B}(hh^\circ h')$ |

Table 1: A summary of notation.

# B Proofs of Presented Results

We next provide proofs of each result from the paper. Our proofs make use of some extra notation: we use $\Rightarrow$ as logical implication, and we use $\mathscr{P}(X)$ to denote the power set of any set $X$. Lastly, we use $\forall_{\mathcal{A} \subseteq X}$ and $\exists_{\mathcal{A} \subseteq X}$ as shorthand for $\forall_{\mathcal{A} \in \mathscr{P}(X)}$ and $\exists_{\mathcal{A} \in \mathscr{P}(X)}$ respectively.

## B.1 Section 3 Proofs

Our first result is from Section 3 of the paper.

**Theorem 3.1.** *For any pair $(\lambda, e)$, there exists infinitely many choices of a basis, $\Lambda_B$, such that both (1) $\lambda \notin \Lambda_B$, and (2) $\Lambda_B \vDash \{\lambda\}$.*

*Proof of Theorem 3.1.*

Choose a fixed but arbitrary pair $(\lambda, e)$. Then, enumerate the realizable histories, $\mathcal{H}^{\lambda,e}$, and let $h^1$ denote the first element of this enumeration, $h^2$ the second, and so on.

Then, we design a constructive procedure for a basis that, when repeatedly applied, induces an infinite enumeration of bases that satisfy the desired two properties. This constructive procedure for the $k$-th basis will contain $k + 1$ agents, where each agent is distinct from $\lambda$, but will produces the same action as the agent every $k + 1$ elements of the history sequence, $h^1, h^2, \ldots$.

For the first ($k = 1$) basis, we construct two agents. The first, $\lambda_B^1$, chooses the same action distribution as $\lambda$ on each even numbered history: $\lambda_B^1(h^i) = \lambda(h^i)$. Then, this agent will choose a different action distribution on the odd length histories: $\lambda_B^1(h^{i+1}) \neq \lambda(h^{i+1})$, for $i$ any even natural number. The second agent, $\lambda_B^2$ will do the opposite to $\lambda_B^1$: on each odd numbered history $h^{i+1}$, $\lambda_B^2(h^{i+1}) \neq \lambda(h^{i+1})$, but on every even numbered history, $\lambda_B^2(h^i) = \lambda(h^i)$.

Observe first that by construction, $\lambda \neq \lambda_B^1$, and $\lambda \neq \lambda_B^2$, since there exist histories where they choose different action distributions. Next, observe that the basis, $\Lambda_B = \{\lambda_B^1, \lambda_B^2\}$, generates $\{\lambda\}$ in $e$ through the following set of learning rules, $\Sigma$: given any realizable history, $h \in \mathcal{H}^{\lambda,e}$, check whether the history has an even or odd numbered index in the enumeration. If odd, choose $\lambda_B^1$, and if even, choose $\lambda_B^2$.

More generally, this procedure can be applied for any $k$:

$$\Lambda_B^k = \{\lambda_B^1, \ldots, \lambda_B^{k+1}\}, \qquad \lambda_B^i(h) = \begin{cases} \lambda(h) & [h] == i, \\ \neq \lambda(h) & \text{otherwise,} \end{cases} \tag{B.1}$$

where we use the notation $[h] == i$ to express the logical predicate asserting that the modulos of the index of $h$ in the enumeration $h^1, h^2, \ldots$ is equal to $i$.

Further, $\neq \lambda(h)$ simply refers to *any* choice of action distribution that is unequal to $\lambda(h)$. Thus, for all natural numbers $k \geq 2$, we can construct a new basis consisting of $k$ base agents that generates $\lambda$ in $e$, but does not contain the agent itself. This completes the argument. $\square$

## B.2 Section 4 Proofs

We next present the proofs of results from Section 4.

### B.2.1 Theorem 4.1: Properties of CRL

We begin with Theorem 4.1 that establishes basic properties of CRL.

**Theorem 4.1.** *Every instance of CRL $(e, v, \Lambda, \Lambda_B)$ satisfies the following properties:*

1. *If $\Lambda \neq \Lambda_B \cup \Lambda^*$, there exists a $\Lambda_B'$ such that (1) $\Lambda_B' \vDash \Lambda$, and (2) $(e, v, \Lambda, \Lambda_B')$ is not an instance of CRL.*

2. *No element of $\Lambda_B$ is optimal: $\Lambda_B \cap \Lambda^* = \emptyset$.*

3. *If $|\Lambda|$ is finite, there exists an agent set, $\Lambda^\circ$ such that $|\Lambda^\circ| < |\Lambda|$ and $\Lambda^\circ \vDash \Lambda$.*

4. *If $|\Lambda|$ is infinite, there exists an agent set, $\Lambda^\circ$ such that $\Lambda^\circ \subset \Lambda$ and $\Lambda^\circ \vDash \Lambda$.*

We prove this result in the form of three lemmas, corresponding to each of the four points of the theorem (with the third lemma, Lemma B.3, covering both points 3. and 4.). Some of the lemmas make use of properties of generates and reaches that we establish later in Appendix C.

**Lemma B.1.** *For all instances of CRL $(e, v, \Lambda, \Lambda_B)$, if $\Lambda \neq \Lambda_B \cup \Lambda^*$, then there exists a choice $\Lambda'_B$ such that (1) $\Lambda'_B \nVDash \Lambda$, and (2) $(e, v, \Lambda, \Lambda'_B)$ is not an instance of CRL.*

*Proof of Lemma B.1.*

Recall that a tuple $(e, v, \Lambda, \Lambda_B)$ is CRL just when all of the optimal agents $\Lambda^*$ do not reach the basis. Then, the result holds as a straightforward consequence of two facts. First, we can always construct a new basis containing all of the optimal agents, $\Lambda_B^\circ = \Lambda_B \cup \Lambda^*$. Notice that $\Lambda_B^\circ$ still generates $\Lambda$ by property three of Theorem 4.2. Further, since both $\Lambda_B$ and $\Lambda^*$ are each subsets of $\Lambda$, and by assumption $\Lambda \neq \Lambda_B \cup \Lambda^*$ (so there is at least one sub-optimal agent that is not in the basis), it follows that $\Lambda_B^\circ \subset \Lambda$. Second, by Proposition C.15, we know that every element $\lambda_B^\circ \in \Lambda_B^\circ$ will always reach the basis, $\lambda_B^\circ \,\square\!\rightsquigarrow \Lambda_B^\circ$. Therefore, in the tuple $(e, v, \Lambda, \Lambda_B^\circ)$, each of the optimal agents will reach the basis, and therefore this is not an instance of CRL. $\square$

**Lemma B.2.** *No element of $\Lambda_B$ is optimal: $\Lambda_B \cap \Lambda^* = \emptyset$.*

*Proof of Lemma B.2.*

The lemma follows as a combination of two facts.

First, recall that, by definition of CRL, each optimal agent $\lambda \in \Lambda^*$ satisfies $\lambda^* \,\nrightsquigarrow \Lambda_B$.

Second, note that by Lemma B.11, we know that each $\lambda_B \in \Lambda_B$ satisfies $\lambda_B \rightsquigarrow \Lambda_B$.

Therefore, since sometimes reaches ($\rightsquigarrow$) and never reaches ($\nrightsquigarrow$) are negations of one another, we conclude that no basis element can be optimal. $\square$

Before stating the next lemma, we note that points (3.) and (4.) of Theorem 4.1 are simply expansions of the definition of a *minimal* agent set, which we define precisely in Definition C.4 and Definition C.5.

**Lemma B.3.** *For any instance of CRL, $\Lambda$ is not minimal.*

*Proof of Lemma B.3.*

We first show that $\Lambda$ cannot be minimal. To do so, we consider the cases where the rank (Definition C.3) of $\Lambda$ is finite and infinite separately.

*(Finite Rank $\Lambda$.)*

If rank$(\Lambda)$ is finite and minimal, then it follows immediately that there is no agent set of smaller rank that generates $\Lambda$. By consequence, since $\Lambda_B \subset \Lambda$ and $\Lambda_B \nVDash \Lambda$, we conclude that $\Lambda$ cannot be minimal. $\checkmark$

*(Infinite Rank $\Lambda$.)*

If rank$(\Lambda)$ is infinite and minimal, then there is no proper subset of $\Lambda$ that uniformly generates $\Lambda$ by definition. By consequence, since $\Lambda_B \subset \Lambda$ and $\Lambda_B \nVDash \Lambda$, we conclude that $\Lambda$ cannot be minimal. $\checkmark$

This completes the argument of both cases, and we conclude that for any instance of CRL, $\Lambda$ is not minimal. $\square$

### B.2.2 Theorem 4.2: Properties of Generates

Next, we prove basic properties of generates.

**Theorem 4.2.** *The following properties hold of the generates operator:*

1. *Generates is transitive: For any triple $(\Lambda^1, \Lambda^2, \Lambda^3)$ and $e \in \mathcal{E}$, if $\Lambda^1 \Vdash^e \Lambda^2$ and $\Lambda^2 \Vdash^e \Lambda^3$, then $\Lambda^1 \Vdash^e \Lambda^3$.*

2. *Generates is not commutative: there exists a pair $(\Lambda^1, \Lambda^2)$ and $e \in \mathcal{E}$ such that $\Lambda^1 \Vdash^e \Lambda^2$, but $\neg(\Lambda^2 \Vdash^e \Lambda^1)$.*

3. *For all $\Lambda$ and pair of agent bases $(\Lambda_B^1, \Lambda_B^2)$ such that $\Lambda_B^1 \subseteq \Lambda_B^2$, if $\Lambda_B^1 \Vdash^e \Lambda$, then $\Lambda_B^2 \Vdash^e \Lambda$.*

4. *For all $\Lambda$ and $e \in \mathcal{E}$, $\Lambda \Vdash^e \Lambda$.*

5. *The decision problem, **Given** $(e, \Lambda_B, \Lambda)$, **output** True iff $\Lambda_B \Vdash^e \Lambda$, is undecidable.*

The proof of this theorem is spread across the next five lemmas below.

The fact that generates is transitive suggests that the basic tools of an agent set—paired with a set of learning rules—might be likened to an algebraic structure. We can draw a symmetry between an agent basis and the basis of a vector space: A vector space is comprised of all linear combinations of the basis, whereas $\Lambda$ is comprised of all valid switches (according to the learning rules) between the base agents. However, the fact that generates is not commutative (by point 2.) raises a natural question: are there choices of learning rules under which generates is commutative? An interesting direction for future work is to explore this style of algebraic analysis on agents.

**Lemma B.4.** *Generates is transitive: For any triple $(\Lambda^1, \Lambda^2, \Lambda^3)$ and $e \in \mathcal{E}$, if $\Lambda^1 \Vdash^e \Lambda^2$ and $\Lambda^2 \Vdash^e \Lambda^3$, then $\Lambda^1 \Vdash^e \Lambda^3$.*

*Proof of Lemma B.4.*

Assume $\Lambda^1 \Vdash^e \Lambda^2$ and $\Lambda^2 \Vdash^e \Lambda^3$. Then, by Proposition C.4 and the definition of the generates operator, we know that

$$\forall_{\lambda^2 \in \Lambda^2} \exists_{\sigma^1 \in \Sigma^1} \forall_{h \in \bar{\mathcal{H}}} \; \lambda^2(h) = \sigma^1(h)(h), \tag{B.2}$$

$$\forall_{\lambda^3 \in \Lambda^3} \exists_{\sigma^2 \in \Sigma^2} \forall_{h \in \bar{\mathcal{H}}} \; \lambda^3(h) = \sigma^2(h)(h), \tag{B.3}$$

where $\Sigma^1$ and $\Sigma^2$ express the set of all learning rules over $\Lambda^1$ and $\Lambda^2$ respectively. By definition of a learning rule, $\sigma$, we rewrite the above as follows,

$$\forall_{\lambda^2 \in \Lambda^2} \forall_{h \in \bar{\mathcal{H}}} \exists_{\lambda^1 \in \Lambda^1} \; \lambda^2(h) = \lambda^1(h), \tag{B.4}$$

$$\forall_{\lambda^3 \in \Lambda^3} \forall_{h \in \bar{\mathcal{H}}} \exists_{\lambda^2 \in \Lambda^2} \; \lambda^3(h) = \lambda^2(h). \tag{B.5}$$

Then, consider a fixed but arbitrary $\lambda^3 \in \Lambda^3$. We construct a learning rule defined over $\Lambda^1$ as $\sigma^1 : \mathcal{H} \to \Lambda^1$ that induces an equivalent agent as follows. For each realizable history, $h \in \bar{\mathcal{H}}$, by Equation B.5 we know that there is an $\lambda^2$ such that $\lambda^3(h) = \lambda^2(h)$, and by Equation B.4, there is an $\lambda^1$ such that $\lambda^2(h) = \lambda^1(h)$. Then, set $\sigma^1 : h \mapsto \lambda^1$ such that $\lambda^1(h) = \lambda^2(h) = \lambda^3(h)$

Since $h$ and $\lambda^3$ were chosen arbitrarily, we conclude that

$$\forall_{\lambda^3 \in \Lambda^3} \forall_{h \in \bar{\mathcal{H}}} \exists_{\lambda^1 \in \Lambda^1} \; \lambda^3(h) = \lambda^1(h).$$

But, by the definition of $\Sigma$, this means there exists a learning rule such that

$$\forall_{\lambda^3 \in \Lambda^3} \exists_{\sigma^1 \in \Sigma^1} \forall_{h \in \bar{\mathcal{H}}} \; \lambda^3(h) = \sigma^1(h)(h).$$

This is exactly the definition of $\Sigma$-generation, and by Proposition C.4, we conclude $\Lambda^1 \Vdash^e \Lambda^3$. $\qquad\square$

**Lemma B.5.** *Generates is not commutative: there exists a pair $(\Lambda^1, \Lambda^2)$ and $e \in \mathcal{E}$ such that $\Lambda^1 \nVdash \Lambda^2$, but $\neg(\Lambda^2 \nVdash \Lambda^1)$.*

**Proof of *Lemma B.5*.**

> The result follows from a simple counterexample: consider the pair
>
> $$\Lambda^1 = \{\lambda_i : h \mapsto a_1\}, \qquad \Lambda^2 = \{\lambda_i : h \mapsto a_1, \lambda_j : h \mapsto a_2\}.$$
>
> Note that since $\lambda_i$ is in both sets, and $\Lambda^1$ is a singleton, we know that $\Lambda^2 \nVdash \Lambda^1$ in any environment. But, by Proposition C.6, we know that $\Lambda^1$ cannot generate $\Lambda^2$. □

**Lemma B.6.** *For all $\Lambda$ and pair of agent bases $(\Lambda_B^1, \Lambda_B^2)$ such that $\Lambda_B^1 \subseteq \Lambda_B^2$, if $\Lambda_B^1 \nVdash \Lambda$, then $\Lambda_B^2 \nVdash \Lambda$.*

**Proof of *Lemma B.6*.**

> The result follows as a natural consequence of the definition of generates. Recall that $\Lambda_B^1 \nVdash \Lambda$ just when,
>
> $$\exists_{\Sigma^1 \subseteq \Sigma} \Lambda_B^1 \nVdash_{\Sigma^1} \Lambda \tag{B.6}$$
>
> $$\equiv \exists_{\Sigma^1 \subseteq \Sigma} \exists_{\sigma^1 \in \Sigma^1} \forall_{h \in \bar{\mathcal{H}}} \lambda(h) = \lambda_B^{\sigma^1(h)}(h), \tag{B.7}$$
>
> where again $\lambda_B^{\sigma^1(h)} \in \Lambda_B^1$ is the base agent chosen by $\sigma^1(h)$. We use superscripts $\Sigma^1$ and $\sigma^1$ to signify that $\sigma^1$ is defined relative to $\Lambda_B^1$, that is, $\sigma^1 : \mathcal{H} \to \Lambda_B^1 \in \Sigma^1$.
>
> But, since $\Lambda_B^1 \subseteq \Lambda_B^2$, we can define $\Sigma^2 = \Sigma^1$ and ensure that $\Lambda_B^2 \nVdash_{\Sigma^2} \Lambda$, since the agent basis $\Lambda_B^1$ was already sufficient to generate $\Lambda$. Therefore, we conclude that $\Lambda_B^2 \nVdash \Lambda$. □

**Lemma B.7.** *For all $\Lambda$ and $e \in \mathcal{E}$, $\mathbb{\Lambda} \nVdash \Lambda$.*

**Proof of *Lemma B.7*.**

> This is a direct consequence of Proposition C.18. □

**Lemma B.8.** *The decision problem,* AGENTSGENERATE, ***Given*** *$(e, \Lambda_B, \Lambda)$,* ***output*** *True iff $\Lambda_B \nVdash \Lambda$, is undecidable.*

**Proof of *Lemma B.8*.**

> We proceed as is typical of such results by reducing AGENTSGENERATE from the Halting Problem.
>
> In particular, let $m$ be a fixed but arbitrary Turing Machine, and $w$ be a fixed but arbitrary input to be given to machine $m$. Then, HALT defines the decision problem that outputs True iff $m$ halts on input $w$.
>
> We construct an oracle for AGENTSGENERATE that can decide HALT as follows. Let $(\mathcal{A}, \mathcal{O})$ be an interface where the observation space is comprised of all configurations of machine $m$. Then, we consider a deterministic environment $e$ that simply produces the next configuration of $m$ when run on input $w$, based on the current tape contents, the state of $m$, and the location of the tape head. Note that all three of these elements are contained in a Turing Machine's configuration, and that a single configuration indicates whether the Turing Machine is in a halting state or not. Now, let the action space $\mathcal{A}$ consist of two actions, $\{a_{\text{no-op}}, a_{\text{halt}}\}$. On execution of $a_{\text{no-op}}$ no-op, the environment moves to the next configuration. On execution of $a_{\text{halt}}$, the machine halts. That is, we restrict ourselves to the singleton agent set, $\Lambda$, containing

the agent $\lambda°$ that outputs $a_{\text{halt}}$ directly following the machine entering a halting configuration, and $a_{\text{no-op}}$ otherwise:

$$\lambda° : hao \mapsto \begin{cases} a_{\text{halt}} & o \text{ is a halting confirmation,} \\ a_{\text{no-op}} & \text{otherwise.} \end{cases}, \qquad \Lambda = \{\lambda°\}.$$

Using these ingredients, we take any instance of HALT, $(m, w)$, and consider the singleton agent basis: $\Lambda_B^1 = \{a_{\text{no-op}}\}$.

We make one query to our AGENTSGENERATE oracle, and ask: $\Lambda_B^1 \nvDash \Lambda$. If it is True, then the histories realizable by $(\lambda°, e)$ pair ensure that the single agent in $\Lambda$ never emits the $a_{\text{halt}}$ action, and thus, $m$ does not halt on $w$. If it is False, then there *are* realizable histories in $e$ in which $m$ halts on $w$. We thus use the oracle's response directly to decide the given instance of HALT. $\qquad\square$

### B.2.3 Theorem 4.3: Properties of Reaches

We find many similar properties hold for reaches.

**Theorem 4.3.** *The following properties hold of the reaches operator:*

1. *$\rightsquigarrow$ and $\nrightsquigarrow$ are not transitive.*

2. *Sometimes reaches is not commutative: there exists a pair $(\Lambda^1, \Lambda^2)$ and $e \in \mathcal{E}$ such that $\forall_{\lambda^1 \in \Lambda^1} \ \lambda^1 \rightsquigarrow \Lambda^2$, but $\exists_{\lambda^2 \in \Lambda^2} \ \lambda^2 \nrightsquigarrow \Lambda^1$.*

3. *For all pairs $(\Lambda, e)$, if $\lambda \in \Lambda$, then $\lambda \rightsquigarrow \Lambda$.*

4. *Every agent satisfies $\lambda \rightsquigarrow \mathbb{A}$ in every environment.*

5. *The decision problem, **Given** $(e, \lambda, \Lambda)$, **output** True iff $\lambda \rightsquigarrow \Lambda$, is undecidable.*

Again, we prove this result through five lemmas that correspond to each of the above properties.

Many of these properties resemble those in Theorem 4.2. For instance, point (5.) shows that deciding whether a given agent sometimes reaches a basis in an environment is undecidable. We anticipate that the majority of decision problems related to determining properties of arbitrary agent sets will be undecidable, though it is still worth making these arguments carefully. Moreover, there may be interesting special cases in which these decision problems are decidable (and perhaps, efficiently so). Identifying these special cases and their corresponding efficient algorithms is another interesting direction for future work.

**Lemma B.9.** *$\rightsquigarrow$ and $\nrightsquigarrow$ are not transitive.*

***Proof of Lemma B.9.***

We construct two counterexamples, one for each of "sometimes reaches" ($\rightsquigarrow$) and "never reaches" ($\nrightsquigarrow$).

**Counterexample: Sometimes Reaches.** To do so, we begin with a tuple $(e, \Lambda^1, \Lambda^2, \Lambda^3)$ such that both

$$\forall_{\Lambda^1 \in \Lambda^1} \lambda^1 \rightsquigarrow \Lambda^1, \qquad \forall_{\Lambda^2 \in \Lambda^2} \lambda^2 \rightsquigarrow \Lambda^2.$$

We will show that there is an agent, $\overline{\lambda}^1 \in \Lambda^1$, such that $\overline{\lambda}^1 \nrightsquigarrow \Lambda^3$, thus illustrating that sometimes reaches is not guaranteed to be transitive. The basic idea is that sometimes reaches only requires an agent stop its search on *one* realizable history. So, $\lambda^1 \rightsquigarrow \Lambda^2$ might happen on some history $h$, but each $\lambda^2 \in \Lambda^2$ might only reach $\Lambda^3$ on an entirely different history. As a result, reaching $\Lambda^2$ is not enough to ensure the agent also reaches $\Lambda^3$.

In more detail, the agent sets of the counterexample are as follows. Let $\mathcal{A} = \{a_1, a_2\}$ and $O = \{o_1, o_2\}$. Let $\Lambda^2$ be all agents that, after ten timesteps, always take $a_2$. $\overline{\lambda}^1$ is simple: it

always takes $a_1$, except on one realizable history, $h^\circ$, (and all of the realizable successors of $h^\circ$, $\mathcal{H}_{h^\circ}^{\overline{\lambda},e}$), where it switches to taking $a_2$ after ten timesteps. Clearly $\overline{\lambda^1} \rightsquigarrow \Lambda^2$, since after ten timesteps, we know there will be some $\lambda^2$ such that $\overline{\lambda}^1(h^\circ h') = \lambda^2(h^\circ h')$ for all realizable history suffixes $h'$. Now, by assumption, we know that $\lambda^2 \rightsquigarrow \Lambda^3$. This ensures there is a *single* realizable history $h$ such that there is an $\lambda^3$ where $\lambda^2(hh') = \lambda^3(hh')$ for any realizable suffix $h'$. To finish the counterexample, we simply note that this realizable $h$ can be different from $h^\circ$ and all of its successors. For example, $h^\circ$ might be the history containing only $o_1$ for the first ten timesteps, while $h$ could be the history containing only $o_2$ for the first ten timesteps. Thus, this $\lambda^1$ never reaches $\Lambda^3$, and we conclude the counterexample. ✓

**Counterexample: Never Reaches.** The instance for never reaches is simple: Let $\mathcal{A} = \{a_1, a_2, a_3\}$, and $\Lambda^1 = \Lambda^3$. Suppose all agents in $\Lambda^1$ (and thus $\Lambda^3$) only choose actions $a_1$ and $a_3$. Let $\Lambda^2$ be a singleton, $\Lambda^2 = \{\lambda^2\}$ such that $\lambda^2 : h \mapsto a_2$. Clearly, every $\lambda^1 \in \Lambda^1$ will never reach $\Lambda^2$, since none of them ever choose $a_2$. Similarly, $\lambda^2$ will never reach $\Lambda^3$, since no agents in $\Lambda^3$ choose $a_2$. However, by [Proposition C.15](#) and the assumption that $\Lambda^1 = \Lambda^3$, we know $\forall_{\lambda^1 \in \Lambda^1} \lambda^1 \square\rightsquigarrow \Lambda^3$. This directly violates transitivity. ✓

This completes the argument for all three cases, and we conclude. □

**Lemma B.10.** *Sometimes reaches is not commutative: there exists a pair $(\Lambda^1, \Lambda^2)$ and $e \in \mathcal{E}$ such that $\forall_{\lambda^1 \in \Lambda^1} \lambda^1 \rightsquigarrow \Lambda^2$, but $\exists_{\lambda^2 \in \Lambda^2} \lambda^2 \not\rightsquigarrow \Lambda^1$.*

*Proof of [Lemma B.10](#).*

The result holds as a straightforward consequence of the following counterexample. Consider the pair of agent sets

$$\Lambda^1 = \{\lambda_i : h \mapsto a_1\}, \qquad \Lambda^2 = \{\lambda_i : h \mapsto a_1, \lambda_j : h \mapsto a_2\}.$$

Note that since $\lambda_i$ is in both sets, and $\Lambda^1$ is a singleton, we know that $\lambda \rightsquigarrow \Lambda^1$ in any environment by [Lemma B.11](#). But, clearly $\lambda_j$ never reaches $\Lambda^1$, since no agent in $\Lambda^1$ *ever* chooses $a_1$. □

**Lemma B.11.** *For all pairs $(\Lambda, e)$, if $\lambda \in \Lambda$, then $\lambda \rightsquigarrow \Lambda$.*

*Proof of [Lemma B.11](#).*

The proposition is straightforward, as any $\lambda \in \Lambda$ will be equivalent to itself in behavior for *all* histories. □

**Lemma B.12.** *Every agent satisfies $\lambda \rightsquigarrow \mathbb{\Lambda}$ in every environment.*

*Proof of [Lemma B.12](#).*

This is again a direct consequence of [Proposition C.18](#). □

**Lemma B.13.** *The decision problem,* AGENTREACHES, *$\textbf{Given}$ $(e, \lambda, \Lambda)$, $\textbf{output}$ True iff $\lambda \rightsquigarrow \Lambda$, is undecidable.*

*Proof of [Lemma B.13](#).*

We again proceed by reducing AGENTREACHES from the Halting Problem.

In particular, let $m$ be a fixed but arbitrary Turing Machine, and $w$ be a fixed but arbitrary input to be given to machine $m$. Then, HALT defines the decision problem that outputs True iff $m$ halts on input $w$.

We construct an oracle for AGENTREACHES that can decide HALT as follows. Consider the same observation space used in the proof of Lemma B.8: Let $\mathcal{O}$ be comprised of all configurations of machine $m$. Then, sequences of observations are simply evolution of different Turing Machines processing possible inputs. We consider an action space, $\mathcal{A} = \{a_{\text{halted}}, a_{\text{not-yet}}\}$, where agents simply report whether the history so far contains a halting configuration.

Then, we consider a deterministic environment $e$ that simply produces the next configuration of $m$ when run on input $w$, based on the current tape contents, the state of $m$, and the location of the tape head. Note again that all three of these elements are contained in a Turing Machine's configuration.

Using these ingredients, we take any instance of HALT, $(m, w)$, and build the singleton agent set $\Lambda_B$ containing only the agent $\lambda_{\text{halted}} : h \mapsto a_{\text{halted}}$ that always reports the machine as having halted. We then consider whether the agent that outputs $a_{\text{not-yet}}$ indefinitely until $m$ reports halting, at which point the agent switches to $a_{\text{halted}}$.

We make one query to our AGENTREACHES oracle, and ask: $\lambda \rightsquigarrow \Lambda_B$. If it is True, then the branching agent eventually becomes equivalent to $\lambda_{\text{halted}}$ in that they both indefinitely output $a_{\text{halted}}$ on at least one realizable history. Since $e$ is deterministic, we know this equivalence holds across all histories. If the query reports False, then there is no future in $e$ in which $m$ halts on $w$, otherwise the agent would become equivalent to $\lambda_{\text{halted}}$. We thus use the oracle's response directly to decide the given instance of HALT. $\qquad\square$

## C  Additional Analysis

Finally, we present a variety of additional results about agents and the generates and reaches operators.

### C.1  Additional Analysis: Generates

We first highlight simple properties of the generates operator. Many of our results build around the notion of *uniform generation*, a variant of the generates operator in which a basis generates an agent set in every environment. We define this operator precisely as follows.

**Definition C.1.** *Let $\Sigma$ be a set of learning rules over some basis $\Lambda_B$. We say that a set $\Lambda$ is **uniformly $\Sigma$-generated** by $\Lambda_B$, denoted $\Lambda_B \models_\Sigma \Lambda$, if and only if*

$$\forall_{\lambda \in \Lambda} \exists_{\sigma \in \Sigma} \forall_{h \in \mathcal{H}} \quad \lambda(h) = \sigma(h)(h). \tag{C.1}$$

**Definition C.2.** *We say a basis $\Lambda_B$ **uniformly generates** $\Lambda$, denoted $\Lambda_B \models \Lambda$, if and only if*

$$\exists_{\Sigma \subseteq \bar{\Sigma}} \quad \Lambda_B \models_\Sigma \Lambda. \tag{C.2}$$

We will first show that uniform generation entails generation in a particular environment. As a consequence, when we prove that certain properties hold of uniform generation, we can typically also conclude that the properties hold for generation as well, though there is some subtlety as to when exactly this implication will allow results about $\models$ to apply directly to $\models^e$.

**Proposition C.1.** *For any $(\Lambda_B, \Lambda)$ pair, if $\Lambda_B \models \Lambda$, then for all $e \in \mathcal{E}$, $\Lambda_B \models^e \Lambda$.*

***Proof of Proposition C.1.***

Recall that in the definition of uniform generation, $\Lambda_B \models \Lambda$, we require,

$$\exists_{\Sigma \subseteq \bar{\Sigma}} \forall_{\lambda \in \Lambda} \exists_{\sigma \in \Sigma} \forall_{h \in \mathcal{H}} \quad \lambda(h) = \sigma(h)(h). \tag{C.3}$$

Now, contrast this with generates with respect to a specific environment $e$,

$$\exists_{\Sigma \subseteq \bar{\Sigma}} \forall_{\lambda \in \Lambda} \exists_{\sigma \in \Sigma} \forall_{h \in \bar{\mathcal{H}}} \quad \lambda(h) = \sigma(h)(h). \tag{C.4}$$

The only difference in the definitions is that the set of histories quantified over is $\mathcal{H}$ in the former, and $\bar{\mathcal{H}} = \mathcal{H}^{\lambda,e}$ in the latter.

Since $\bar{\mathcal{H}} \subseteq \mathcal{H}$ for any choice of environment $e$, we can conclude that when Equation C.3, it is also the case that Equation C.4 holds, too. Therefore, $\Lambda_B \models \Lambda \implies \Lambda_B \nvDash \Lambda$ for any $e$. $\quad\square$

We next show that the subset relation implies generation.

**Proposition C.2.** *Any pair of agent sets* $(\Lambda_{\text{small}}, \Lambda_{\text{big}})$ *such that* $\Lambda_{\text{small}} \subseteq \Lambda_{\text{big}}$ *satisfies*

$$\Lambda_{\text{big}} \models \Lambda_{\text{small}}. \tag{C.5}$$

*Proof of Proposition C.2.*

The result follows from the combination of two facts. First, that all agent sets generate themselves. That is, for arbitrary $\Lambda$, we know that $\Lambda \nvDash \Lambda$, since the trivial set of learning rules,

$$\Sigma_{\text{tr}} = \{\sigma_i : h \mapsto \lambda_i, \ \forall_{\lambda_i \in \Lambda}\}, \tag{C.6}$$

that never switches between agents is sufficient to generate the agent set.

Second, observe that removing an agent from the generated set has no effect on the generates operator. That is, let $\Lambda' = \Lambda \setminus \lambda$, for fixed but arbitrary $\lambda \in \Lambda$. We see that $\Lambda \nvDash \Lambda'$, since $\Sigma_{\text{tr}}$ is sufficient to generate $\Lambda'$, too. By inducting over all removals of agents from $\Lambda$, we reach our conclusion. $\quad\square$

Next, we establish properties about the sets of learning rules that correspond to the generates operator.

**Proposition C.3.** *For any* $(\Lambda_B, \Sigma, \Lambda)$ *such that* $\Lambda_B \models_\Sigma \Lambda$, *it holds that*

$$|\Lambda| \le |\Sigma|. \tag{C.7}$$

*Proof of Proposition C.3.*

We proceed toward contradiction, and assume $|\Lambda| > |\Sigma|$. Then, there is at least one learning rule $\sigma \in \Sigma$ that corresponds to two or more distinct agents in $\Lambda$. Call this element $\sigma^\circ$, and without loss of generality let $\lambda^1$ and $\lambda^2$ be two distinct agents that are each generated by $\sigma^\circ$ in the sense that,
$$\lambda^1(h) = \sigma^\circ(h)(h), \qquad \lambda^2(h) = \sigma^\circ(h)(h), \tag{C.8}$$
for every $h \in \mathcal{H}$. But, by the distinctness of $\lambda^1$ and $\lambda^2$, there must exist a history $h$ in which $\lambda^1(h) \ne \lambda^2(h)$. We now arrive at a contradiction as such a history cannot exist: By Equation C.8, we know that $\lambda^1(h) = \sigma^\circ(h)(h) = \lambda^2(h)$ for all $h$. $\quad\square$

We see that the universal learning rules, $\bar{\Sigma}$, is the strongest in the following sense.

**Proposition C.4.** *For any basis* $\Lambda_B$ *and agent set* $\Lambda$, *exactly one of the two following properties hold:*

    *1. The agent basis* $\Lambda_B$ *uniformly generates* $\Lambda$ *under the set of all learning rules:* $\Lambda_B \models_{\bar{\Sigma}} \Lambda$.

    *2. There is no set of learning rules for which the basis* $\Sigma$*-uniformly generates the agent set:* $\neg \exists_{\Sigma \subseteq \bar{\Sigma}} \ \Lambda_B \models_\Sigma \Lambda$.

*Proof of Proposition C.4.*

The proof follows from the law of excluded middle. That is, for any set of learning rules $\Sigma$, either it generates $\Lambda$ or it does not. If it does generate $\Lambda$, by Lemma B.6 so does $\bar{\Sigma}$. By consequence, if $\bar{\Sigma}$ does *not* generate $\Lambda$, neither do any of its subsets. $\quad\square$

Furthermore, uniform generation is also transitive.

**Theorem C.5.** *Uniform generates is transitive: For any triple $(\Lambda^1, \Lambda^2, \Lambda^3)$, if $\Lambda^1 \models \Lambda^2$ and $\Lambda^2 \models \Lambda^3$, then $\Lambda^1 \models \Lambda^3$.*

*Proof of Theorem C.5.*

Assume $\Lambda^1 \models \Lambda^2$ and $\Lambda^2 \models \Lambda^3$. Then, by Proposition C.4 and the definition of the uniform generates operator, we know that

$$\forall_{\lambda^2 \in \Lambda^2} \exists_{\sigma^1 \in \Sigma^1} \forall_{h \in \mathcal{H}} \; \lambda^2(h) = \sigma^1(h)(h), \tag{C.9}$$

$$\forall_{\lambda^3 \in \Lambda^3} \exists_{\sigma^2 \in \Sigma^2} \forall_{h \in \mathcal{H}} \; \lambda^3(h) = \sigma^2(h)(h), \tag{C.10}$$

where $\Sigma^1$ and $\Sigma^2$ express the set of all learning rules over $\Lambda^1$ and $\Lambda^2$ respectively. By definition of a learning rule, $\sigma$, we rewrite the above as follows,

$$\forall_{\lambda^2 \in \Lambda^2} \forall_{h \in \mathcal{H}} \exists_{\lambda^1 \in \Lambda^1} \; \lambda^2(h) = \lambda^1(h), \tag{C.11}$$

$$\forall_{\lambda^3 \in \Lambda^3} \forall_{h \in \mathcal{H}} \exists_{\lambda^2 \in \Lambda^2} \; \lambda^3(h) = \lambda^2(h). \tag{C.12}$$

Then, consider a fixed but arbitrary $\lambda^3 \in \Lambda^3$. We construct a learning rule defined over $\Lambda^1$ as $\sigma^1 : \mathcal{H} \to \Lambda^1$ that induces an equivalent agent as follows. For each history, $h \in \mathcal{H}$, by Equation C.12 we know that there is an $\lambda^2$ such that $\lambda^3(h) = \lambda^2(h)$, and by Equation C.11, there is an $\lambda^1$ such that $\lambda^2(h) = \lambda^1(h)$. Then, set $\sigma^1 : h \mapsto \lambda^1$ such that $\lambda^1(h) = \lambda^2(h) = \lambda^3(h)$. Since $h$ and $\lambda^3$ were chosen arbitrarily, we conclude that

$$\forall_{\lambda^3 \in \Lambda^3} \forall_{h \in \mathcal{H}} \exists_{\lambda^1 \in \Lambda^1} \; \lambda^3(h) = \lambda^1(h).$$

But, by the definition of $\Sigma$, this means there exists a learning rule such that

$$\forall_{\lambda^3 \in \Lambda^3} \exists_{\sigma^1 \in \Sigma^1} \forall_{h \in \mathcal{H}} \; \lambda^3(h) = \sigma^1(h)(h).$$

This is exactly the definition of $\Sigma$-uniform generation, and by Proposition C.4, we conclude $\Lambda^1 \models \Lambda^3$. $\qquad\square$

Next, we show that a singleton basis only generates itself.

**Proposition C.6.** *Any singleton basis, $\Lambda_B = \{\lambda\}$, only uniformly generates itself.*

*Proof of Proposition C.6.*

Note that generates requires switching between base agents. With only a single agent, there cannot be any switching, and thus, the only agent that can be described as switching amongst the elements of the singleton set $\Lambda_B = \{\lambda\}$ is the set itself. $\qquad\square$

### C.1.1 Rank and Minimal Bases

As discussed in the paper, one natural reaction to the concept of an agent basis is to ask how we can justify different choices of a basis. And, if we cannot, then perhaps the concept of an agent basis is disruptive, rather than illuminating. In the main text, we suggest that in many situations, the choice of basis is made by the constraints imposed by the problem, such as the available memory. However, there are some objective properties of different bases that can help us to evaluate possible choices of a suitable basis. For instance, some bases are minimal in the sense that they cannot be made smaller while still retaining the same expressive power (that is, while generating the same agent sets). Identifying such minimal sets may be useful, as it is likely that there is good reason to consider only the most compressed agent bases.

To make these intuitions concrete, we introduce the *rank* of an agent set.

**Definition C.3.** *The **rank** of an agent set,* $\mathrm{rank}(\Lambda)$*, is the size of the smallest agent basis that uniformly generates it:*

$$\mathrm{rank}(\Lambda) = \min_{\Lambda_B \subset \mathbb{\Lambda}} |\Lambda_B| \qquad s.t. \qquad \Lambda_B \models \Lambda. \tag{C.13}$$

For example, the agent set,

$$\Lambda = \{\lambda^0 : h \mapsto a_0, \tag{C.14}$$
$$\lambda^1 : h \mapsto a_1,$$
$$\lambda^2 : h \mapsto \begin{cases} a_0 & |h| \bmod 2 = 0, \\ a_1 & |h| \bmod 2 = 1, \end{cases}$$
$$\},$$

has $\mathrm{rank}(\Lambda) = 2$, since the basis,

$$\Lambda_{\mathrm{B}} = \{\lambda_{\mathrm{B}}^0 : h \mapsto a_0, \ \lambda_{\mathrm{B}}^1 : h \mapsto a_1\},$$

uniformly generates $\Lambda$, and there is no size-one basis that uniformly generates $\Lambda$ by Proposition C.6.

Using the notion of an agent set's rank, we now introduce the concept of a *minimal basis*. We suggest that minimal bases are particular important, as they contain no redundancy with respect to their expressive power. Concretely, we define a minimal basis in two slightly different ways depending on whether the basis has finite or infinite rank. In the finite case, we say a basis is minimal if there is no basis of lower rank that generates it.

**Definition C.4.** *An agent basis $\Lambda_{\mathrm{B}}$ with finite rank is said to be **minimal** just when there is no smaller basis that generates it,*

$$\forall_{\Lambda_{\mathrm{B}}' \subset \Lambda} \ \Lambda_{\mathrm{B}}' \models \Lambda_{\mathrm{B}} \implies \mathrm{rank}(\Lambda_{\mathrm{B}}') \geq \mathrm{rank}(\Lambda_{\mathrm{B}}). \tag{C.15}$$

In the infinite case, as all infinite rank bases will have the same effective size, we instead consider a notion of minimiality based on whether any elements can be removed from the basis without changing its expressive power.

**Definition C.5.** *An agent basis $\Lambda_{\mathrm{B}}$ with infinite rank is said to be **minimal** just when no proper subset of $\Lambda_{\mathrm{B}}$ uniformly generates $\Lambda_{\mathrm{B}}$.*

$$\forall_{\Lambda_{\mathrm{B}}' \subseteq \Lambda_{\mathrm{B}}} \ \Lambda_{\mathrm{B}}' \models \Lambda_{\mathrm{B}} \implies \Lambda_{\mathrm{B}}' = \Lambda_{\mathrm{B}}. \tag{C.16}$$

Notably, this way of looking at minimal bases will also apply to finite rank agent bases as a direct consequence of the definition of a minimal finite rank basis. However, we still provide both definitions, as a finite rank basis may not contain a subset that generates it, but there may exist a lower rank basis that generates it.

**Corollary C.7.** *As a Corollary of Proposition C.2 and Definition C.4, for any minimal agent basis $\Lambda_{\mathrm{B}}$, there is no proper subset of $\Lambda_{\mathrm{B}}$ that generates $\Lambda_{\mathrm{B}}$.*

Regardless of whether an agent basis has finite or infinite rank, we say the basis is a minimal basis of an agent set $\Lambda$ just when the basis uniformly generates $\Lambda$ and the basis is minimal.

**Definition C.6.** *For any $\Lambda$, a **minimal basis of** $\Lambda$ is any basis $\Lambda_{\mathrm{B}}$ that is both (1) minimal, and (2) $\Lambda_{\mathrm{B}} \models \Lambda$.*

A natural question arises as to whether the minimal basis of any agent set $\Lambda$ is unique. We answer this question in the negative.

**Proposition C.8.** *The minimal basis of a set of agents is not necessarily unique.*

***Proof of Proposition C.8.***

To prove the claim, we construct an instance of an agent set with two distinct minimal bases.

Let $\mathcal{A} = \{a_0, a_1\}$, and $O = \{o_0\}$. We consider the agent set containing four agents. The first two map every history to $a_0$ and $a_1$, respectively, while the second two alternate between $a_0$

and $a_1$ depending on whether the history is of odd or even length:

$$\Lambda = \{\lambda^0 : h \mapsto a_0, \qquad (C.17)$$

$$\lambda^1 : h \mapsto a_1,$$

$$\lambda^2 : h \mapsto \begin{cases} a_0 & |h| \bmod 2 = 0, \\ a_1 & |h| \bmod 2 = 1, \end{cases}$$

$$\lambda^3 : h \mapsto \begin{cases} a_0 & |h| \bmod 2 = 1, \\ a_1 & |h| \bmod 2 = 0, \end{cases}$$

$$\}.$$

Note that there are two distinct subsets that each universally generate $\Lambda$:

$$\Lambda_B^{0,1} = \{\lambda_B^0, \lambda_B^1\}, \qquad \Lambda_B^{2,3} = \{\lambda_B^2, \lambda_B^3\}. \qquad (C.18)$$

Next notice that there cannot be a singleton basis by Proposition C.6, and thus, both $\Lambda_B^{0,1}$ and $\Lambda_B^{2,3}$ satisfy (1) $|\Lambda_B^{0,1}| = |\Lambda_B^{2,3}| = \text{rank}(\Lambda)$, and (2) both $\Lambda_B^{0,1} \models \Lambda$, $\Lambda_B^{2,3} \models \Lambda$. $\qquad \square$

Beyond the lack of redundancy of a basis, we may also be interested in their expressive power. For instance, if we compare two minimal bases, $\Lambda_B^1$ and $\Lambda_B^2$, how might we justify which to use? To address this question, we consider another desirable property of a basis: *universality*.

**Definition C.7.** *An agent basis $\Lambda_B$ is **universal** if $\Lambda_B \models \mathbb{A}$.*

Clearly, it might be desirable to work with a universal basis, as doing so ensures that the set of agents we consider in our design space is as rich as possible. We next show that there is at least one natural basis that is both minimal and universal.

**Proposition C.9.** *The basis,*

$$\Lambda_B^\circ = \{\lambda : O \rightarrow \Delta(\mathcal{A})\}, \qquad (C.19)$$

*is a **minimal universal basis**:*

1. *$\Lambda_B^\circ \models \mathbb{A}$: The basis uniformly generates the set of all agents.*

2. *$\Lambda_B^\circ$ is minimal.*

***Proof of Proposition C.9.***

We prove each property separately.

*1. $\Lambda_B^\circ \models \Lambda$*

First, we show that the basis is universal: $\Lambda_B^\circ \models \mathbb{A}$. Recall that this amounts to showing that,

$$\forall_{\lambda \in \mathbb{A}} \forall_{h \in \mathcal{H}} \exists_{\lambda' \in \Lambda_B^\circ} \; \lambda(h) = \lambda'(h). \qquad (C.20)$$

Let $\lambda \in \mathbb{A}$ and $h \in \mathcal{H}$ be fixed but arbitrary. Now, let us label the action distribution produced by $\lambda(h)$ as $p_{\lambda(h)}$. Let $o$ refer to the last observation contained in $h$ (or $\emptyset$ if $h = h_0 = \emptyset$). Now, construct the agent $\lambda_B^\circ : o \mapsto p_{\lambda(h)}$. By construction of $\Lambda_B^\circ$, this agent is guaranteed to be a member of $\Lambda_B^\circ$, and furthermore, we know that $\lambda_B^\circ$ produces the same output as $\lambda$ on $h$. Since both $\lambda$ and $h$ were chosen arbitrarily, the construction will work for any choice of $\lambda$ and $h$, and we conclude that at every history, there exists a basis agent $\Lambda_B^\circ \in \Lambda_B^\circ$ that produces the same probability distribution over actions as any given agent. Thus, the first property holds. $\checkmark$

*2. $\Lambda_B^\circ$ is minimal.*

Second, we show that $\Lambda_B^\circ$ is a minimal basis of $\mathbb{A}$. Recall that since $\text{rank}(\Lambda_B^\circ) = \infty$, the definition of a minimal basis means that:

$$\forall_{\Lambda_B \subseteq \Lambda_B^\circ} \ \Lambda_B \models \mathbb{A} \implies \Lambda_B = \Lambda_B^\circ. \tag{C.21}$$

To do so, fix an arbitrary proper subset of $\Lambda_B \in \mathscr{P}(\Lambda_B^\circ)$. Notice that since $\Lambda_B$ is a proper subset, there exists a non-empty set $\overline{\Lambda_B}$ such that,

$$\Lambda_B \cup \overline{\Lambda_B} = \Lambda_B^\circ.$$

Now, we show that $\Lambda_B$ cannot uniformly generate $\mathbb{A}$ by constructing an agent from $\overline{\Lambda_B}$. In particular, consider the first element of $\overline{\Lambda_B}$, which, by construction of $\Lambda_B^\circ$, is *some* mapping from $O$ to a choice of probability distribution over $\mathcal{A}$. Let us refer to this agent's output probability distribution over actions as $\overline{p}$. Notice that there cannot exist an agent in $\Lambda_B^\circ$ that chooses $\overline{p}$, otherwise $\Lambda_B$ would not be a proper subset of $\Lambda_B^\circ$. Notice further that in the set of all agents, there are infinitely many agents that output $\overline{p}$ in at least one history. We conclude that $\Lambda_B$ cannot uniformly generate $\mathbb{A}$, as it does not contain any base element that produces $\overline{p}$. The set $\overline{\Lambda_B}$ was chosen arbitrarily, and thus the claim holds for any proper subset of $\Lambda_B^\circ$, and we conclude. $\checkmark$

This completes the proof of both statements. $\square$

**Corollary C.10.** *As a direct consequence of Proposition C.9, every universal basis has infinite rank.*

### C.1.2 Orthogonal and Parallel Agent Sets

Drawing inspiration from vector spaces, we introduce notions of *orthogonal* and *parallel* agent bases according to the agent sets they generate.

**Definition C.8.** *A pair of agent bases $(\Lambda_B^1, \Lambda_B^2)$ are **orthogonal** if any pair $(\Lambda^1, \Lambda^2)$ they each uniformly generate*

$$\Lambda_B^1 \models \Lambda^1, \qquad \Lambda_B^2 \models \Lambda^2, \tag{C.22}$$

*satisfy*

$$\Lambda^1 \cap \Lambda^2 = \emptyset. \tag{C.23}$$

Naturally this definition can be modified to account for environment-relative generation ($\models^{\natural}$), or to be defined with respect to a particular set of learning rules in which case two bases are orthogonal with respect to the learning rule set just when they generate different agent sets under the given learning rules. As with the variants of the two operators, we believe the details of such formalisms are easy to produce.

A few properties hold of any pair of orthogonal bases.

**Proposition C.11.** *If two bases $\Lambda_B^1, \Lambda_B^2$ are orthogonal, then the following properties hold:*

1. $\Lambda_B^1 \cap \Lambda_B^2 = \emptyset$.

2. *Neither $\Lambda_B^1$ nor $\Lambda_B^2$ are universal.*

***Proof of Proposition C.11.***

We prove each property independently.

*1. $\Lambda_B^1 \cap \Lambda_B^2 = \emptyset$*

We proceed toward contradiction. That is, suppose that both $\Lambda_B^1$ is orthogonal to $\Lambda_B^2$, and that $\Lambda_B^1 \cap \Lambda_B^2 \neq \emptyset$. Then, by the latter property, there is at least one agent that is an element of

both bases. Call this agent $\lambda_B^\circ$. It follows that the set $\Lambda_B^\circ = \{\lambda_B^\circ\}$ is a subset of both $\Lambda_B^1$ and $\Lambda_B^2$. By Proposition C.2, it follows that $\Lambda_B^1 \models \Lambda_B^\circ$ and $\Lambda_B^2 \models \Lambda_B^\circ$. But this contradicts the fact that $\Lambda_B^1$ is orthogonal to $\Lambda_B^2$, and so we conclude. $\checkmark$

*2. Neither $\Lambda_B^1$ nor $\Lambda_B^2$ are universal.*

We again proceed toward contradiction. Suppose without loss of generality that $\Lambda_B^1$ is universal. Then, we know $\Lambda_B^1 \models \mathbb{A}$. Now, we consider two cases: either $\Lambda_B^2$ generates some non-empty set, $\Lambda^2$, or it does not generate any sets. If it generates a set $\Lambda^2$, then we arrive at a contradiction as $\Lambda^2 \cap \mathbb{A} \neq \emptyset$, which violates the definition of orthogonal bases. If if does not generate a set, this violates the definition of a basis, as any basis is by construction non-empty, and we know that containing even a single element is sufficient to generate at least one agent set by Proposition C.6. Therefore, in either of the two cases, we arrive at a contradiction, and thus conclude the argument. $\checkmark$

This concludes the proof of each statement. $\qquad\qquad\qquad\qquad\qquad\qquad\qquad\qquad$ $\square$

**Corollary C.12.** *For any non-universal agent basis $\Lambda_B$, there exists an orthogonal agent basis, $\Lambda_B^\dagger$.*

Conversely, two agent bases $\Lambda_B^1, \Lambda_B^2$ are parallel just when they generate the same agent sets.

**Definition C.9.** *A pair of agent bases $(\Lambda_B^1, \Lambda_B^2)$ are **parallel** if for every agent set $\Lambda$, $\Lambda_B^1 \models \Lambda$ if and only if $\Lambda_B^2 \models \Lambda$.*

**Proposition C.13.** *If two bases $\Lambda_B^1, \Lambda_B^2$ are parallel, then the following properties hold:*

1. *Both $\Lambda_B^1 \models \Lambda_B^2$ and $\Lambda_B^2 \models \Lambda_B^1$.*

2. $\mathrm{rank}(\Lambda_B^1) = \mathrm{rank}(\Lambda_B^2)$.

3. *$\Lambda_B^1$ is universal if and only if $\Lambda_B^2$ is universal.*

***Proof of Proposition C.13.***

We prove each property separately.

*1. Both $\Lambda_B^1 \models \Lambda_B^2$ and $\Lambda_B^2 \models \Lambda_B^1$.*

The claim follows directly from the definition of parallel bases. An agent set $\Lambda$ is uniformly generated by $\Lambda_B^1$ if and only if it is uniformly generated by $\Lambda_B^2$. Since by Proposition C.2 we know both $\Lambda_B^1 \models \Lambda_B^1$ and $\Lambda_B^2 \models \Lambda_B^2$, we conclude that both $\Lambda_B^1 \models \Lambda_B^2$ and $\Lambda_B^2 \models \Lambda_B^1$. $\checkmark$

*2. $\mathrm{rank}(\Lambda_B^1) = \mathrm{rank}(\Lambda_B^2)$.*

Recall that the definition of rank refers to the size of the smallest basis that uniformly generates it,

$$\mathrm{rank}(\Lambda) = \min_{\Lambda_B \subset \mathbb{A}} |\Lambda_B| \qquad \text{s.t.} \qquad \Lambda_B \models \Lambda.$$

Now, note that by property (1.) of the proposition, both sets uniformly generate each other. Therefore, we know that

$$\mathrm{rank}(\Lambda_B^1) \leq \min\left\{|\Lambda_B^1|, |\Lambda_B^2|\right\}, \qquad \mathrm{rank}(\Lambda_B^2) \leq \min\left\{|\Lambda_B^1|, |\Lambda_B^2|\right\},$$

since the smallest set that generates each basis is no larger than the basis itself, or the other basis.

*3. $\Lambda_{\mathrm{B}}^1$ is universal if and only if $\Lambda_{\mathrm{B}}^2$ is universal.*

The claim again follows by combining the definitions of universality and parallel: If $\Lambda_{\mathrm{B}}^1$ is universal, then by definition of parallel bases, $\Lambda_{\mathrm{B}}^2$ must uniformly generate all the same agent sets including $\mathbb{A}$, and therefore $\Lambda_{\mathrm{B}}^2$ is universal, too. Now, if $\Lambda_{\mathrm{B}}^1$ is not universal, then it does not uniformly generate $\mathbb{A}$. By the definition of parallel bases, we conclude $\Lambda_{\mathrm{B}}^2$ does not generate $\mathbb{A}$ as well. Both directions hold for each labeling of the two bases without loss of generality, and we conclude. ✓

This completes the argument for each property, and we conclude. □

## C.2  Analysis: Reaches

We now establish other properties of the reaches operator. Several of these results are based on a third modality of the reaches operator: *always reaches*, in which an agent eventually reaches an agent basis in *all* histories realizable in a given environment. We define this precisely as follows.

**Definition C.10.** *We say agent $\lambda \in \mathbb{A}$ **always reaches** $\Lambda_{\mathrm{B}}$, denoted $\lambda \mathrel{\Box\leadsto} \Lambda_{\mathrm{B}}$, if and only if*

$$\forall_{h\in\bar{\mathcal{H}}} \exists_{t\in\mathbb{N}_0} \forall_{h^\circ\in\bar{\mathcal{H}}_h^{t:\infty}} \exists_{\lambda_{\mathrm{B}}\in\Lambda_{\mathrm{B}}} \forall_{h'\in\bar{\mathcal{H}}_h} \; \lambda(hh^\circ h') = \lambda_{\mathrm{B}}(hh^\circ h'). \tag{C.24}$$

The nested quantifiers allows the agent to become equivalent to *different* base behaviors depending on the evolution of the interaction stream. For example, in an environment that flips a coin to determine whether $a_{\text{heads}}$ or $a_{\text{tails}}$ is optimal, the $\lambda$ might output $a_{\text{heads}}$ indefinitely if the coin is heads, but $a_{\text{tails}}$ otherwise. In this case, such an agent will still always reach the basis $\Lambda_{\mathrm{B}} = \{\lambda_{\mathrm{B}}^1 : h \mapsto a_{\text{heads}}, \lambda_{\mathrm{B}}^2 : h \mapsto a_{\text{tails}}\}$. Notice that we here make use of the notation, $\bar{\mathcal{H}}_h^{t:\infty}$, which refers to all history suffixes of length $t$ or greater, defined precisely as

$$\bar{\mathcal{H}}_h^{t:\infty} = \{h' \in \bar{\mathcal{H}}_h : |h'| \geq t\} \tag{C.25}$$

We first show that the *always* reaches operator implies *sometimes* reaches.

**Proposition C.14.** *If $\lambda \mathrel{\Box\leadsto} \Lambda$, then $\lambda \leadsto \Lambda$.*

***Proof of Proposition C.14.***

Assume $\lambda \mathrel{\Box\leadsto} \Lambda$. That is, expanding the definition of always reaches, we assume

$$\forall_{h\in\bar{\mathcal{H}}} \exists_{t\in\mathbb{N}_0} \forall_{h^\circ\in\bar{\mathcal{H}}_h^{t:\infty}} \exists_{\lambda_{\mathrm{B}}\in\Lambda_{\mathrm{B}}} \forall_{h'\in\bar{\mathcal{H}}_h} \; \lambda(hh^\circ h') = \lambda_{\mathrm{B}}(hh^\circ h'). \tag{C.26}$$

Further recall the definition of can reach $\lambda \leadsto \Lambda_{\mathrm{B}}$ is as follows

$$\exists_{h\in\bar{\mathcal{H}}} \exists_{\lambda_{\mathrm{B}}\in\Lambda_{\mathrm{B}}} \forall_{h'\in\bar{\mathcal{H}}_h} \; \lambda(hh') = \lambda_{\mathrm{B}}(hh'). \tag{C.27}$$

Then, the claim follows quite naturally: pick any realizable history $h \in \bar{\mathcal{H}}$. By our initial assumption that $\lambda \mathrel{\Box\leadsto} \Lambda$, it follows (by Equation C.26) that there is a time $t$ and a realizable history suffix $h^\circ$ for which

$$\exists_{\lambda_{\mathrm{B}}\in\Lambda_{\mathrm{B}}} \forall_{h'\in\bar{\mathcal{H}}_h} \; \lambda(hh^\circ h') = \lambda_{\mathrm{B}}(hh^\circ h').$$

By construction of $hh^\circ \in \bar{\mathcal{H}}_h$, we know $hh^\circ$ is a realizable history. Therefore, there exists a realizable history, $h^* = hh^\circ$, for which $\exists_{\lambda_{\mathrm{B}}\in\Lambda_{\mathrm{B}}} \forall_{h'\in\bar{\mathcal{H}}_h} \; \lambda(h^*h') = \lambda_{\mathrm{B}}(h^*h')$ holds. But this is exactly the definition of can reach, and therefore, we conclude the argument. □

Next, we highlight the fact that every agent in a basis also reaches that basis.

**Proposition C.15.** *For any agent set $\Lambda$, it holds that $\lambda \mathrel{\Box\leadsto} \Lambda$ for every $\lambda \in \Lambda$.*

***Proof of Proposition C.15.***

The proposition is straightforward, as any $\lambda \in \Lambda$ will be equivalent to itself in behavior for *all* histories. $\quad\square$

**Corollary C.16.** *As a corollary of [Proposition C.15](#), any pair of agent sets* $(\Lambda_{\text{small}}, \Lambda_{\text{big}})$ *where* $\Lambda_{\text{small}} \subseteq \Lambda_{\text{big}}$, *satisfies*

$$\forall_{\lambda \in \Lambda_{\text{small}}} \quad \lambda \; \Box\!\leadsto \; \Lambda_{\text{big}}. \tag{C.28}$$

We further show that, unlike sometimes and never reaches, always reaches is transitive.

**Proposition C.17.** *Always reaches is transitive.*

*Proof of [Proposition C.17](#).*

We proceed by assuming that both $\forall_{\lambda^1 \in \Lambda^1} \; \lambda^1 \; \Box\!\leadsto \; \Lambda^2$ and $\forall_{\lambda^2 \in \Lambda^2} \; \lambda^2 \; \Box\!\leadsto \; \Lambda^3$ and show that it must follow that $\forall_{\lambda^1 \in \Lambda^1} \; \lambda^1 \; \Box\!\leadsto \; \Lambda^3$. To do so, pick a fixed but arbitrary $\lambda^1 \in \Lambda^1$, and expand $\lambda^1 \; \Box\!\leadsto \; \Lambda^2$ as

$$\forall_{h \in \bar{\mathcal{H}}} \exists_{t \in \mathbb{N}_0} \forall_{h^\circ \in \bar{\mathcal{H}}_h^{t:\infty}} \exists_{\lambda^2 \in \Lambda^2} \forall_{h' \in \bar{\mathcal{H}}_h} \; \lambda^1(hh^\circ h') = \lambda^2(hh^\circ h').$$

Now, consider for any realizable history $hh^\circ h'$, we know that the corresponding $\lambda^2$ that produces the same action distribution as $\lambda^1$ also satisfies $\lambda^2 \; \Box\!\leadsto \; \Lambda^3$. Thus, there must exist *some* time $\bar{t}$ at which, any realizable history $\bar{h}\bar{h}^\circ$, will satisfy $\exists_{\lambda^3 \in \Lambda^3} \forall_{\bar{h}' \in \bar{\mathcal{H}}} \; \lambda^2(\bar{h}\bar{h}^\circ \bar{h}') = \lambda^3(\bar{h}\bar{h}^\circ \bar{h}')$. But then there exists a time $(\bar{t})$, that ensures every $\lambda^2 \in \Lambda^2$ will have a corresponding $\lambda^3 \in \Lambda^3$ with the same action distribution at all subsequent realizable histories.

Therefore,

$$\forall_{h \in \bar{\mathcal{H}}} \exists_{t' \in \mathbb{N}_0} \forall_{h^\circ \in \bar{\mathcal{H}}_{ht':\infty}} \exists_{\lambda^2 \in \Lambda^2} \forall_{h' \in \bar{\mathcal{H}}_h} \; \lambda^1(hh^\circ h') = \underbrace{\lambda^2(hh^\circ h')}_{\exists_{\lambda^3 \in \Lambda^3} \; = \lambda^3(hh^\circ h')} .$$

Thus, rewriting,

$$\forall_{h \in \bar{\mathcal{H}}} \exists_{t' \in \mathbb{N}_0} \forall_{h^\circ \in \bar{\mathcal{H}}_h^{t':\infty}} \exists_{\lambda^3 \in \Lambda^3} \forall_{h' \in \bar{\mathcal{H}}_h} \; \lambda^1(hh^\circ h') = \lambda^3(hh^\circ h').$$

But this is precisely the definition of always reaches, and thus we conclude. $\quad\square$

Next, we show two basic properties of the set of all agents: it uniformly generates all agent sets, and it is always reached by all agents.

**Proposition C.18.** *For any $e$, the set of all agents $\mathbb{\Lambda}$ (i) uniformly generates all other agent sets, and (ii) is always reached by all agents:*

$$(i) \quad \forall_{\Lambda \subseteq \mathbb{\Lambda}} \; \mathbb{\Lambda} \models \Lambda, \qquad\qquad (ii) \quad \forall_{\lambda \in \mathbb{\Lambda}} \; \lambda \; \Box\!\leadsto \; \mathbb{\Lambda}. \tag{C.29}$$

*Proof of [Proposition C.18](#).*

*(i).* $\forall_{\Lambda \subseteq \mathbb{\Lambda}} \; \mathbb{\Lambda} \models \Lambda$

The property holds as a straightforward consequence of [Proposition C.2](#): Since any set $\Lambda$ is a subset of $\mathbb{\Lambda}$, it follows that $\mathbb{\Lambda} \models \Lambda$. $\checkmark$

*(ii).* $\forall_{\lambda \in \mathbb{\Lambda}} \; \lambda \; \Box\!\leadsto \; \mathbb{\Lambda}$

The property holds as a straightforward consequence of [Proposition C.15](#): Since every agent satisfies $\lambda \in \mathbb{\Lambda}$, it follows that $\lambda \leadsto \mathbb{\Lambda}$. $\checkmark$

This concludes the argument of both statements. $\quad\square$

## C.3 Figure: Set Relations in CRL

Finally, in Figure 3 we present a visual depicting the set relations in CRL between an agent basis $\Lambda_B$, an agent set it generates $\Lambda$, and the three agent sets corresponding to those agents in $\Lambda$ that (i) sometimes, (ii) never, or (iii) always reach the basis. First, we highlight that we visualize $\Lambda_B$ as a subset of $\Lambda$ since we define $\Lambda_B \subset \Lambda$ in CRL (Definition 4.2). However, there can exist triples $(\Lambda_B, \Lambda, e)$ such that $\Lambda_B \not\models \Lambda$, but that $\Lambda_B$ is *not* a subset of $\Lambda$. Such cases are slightly peculiar, since it means that the basis contains agents that cannot be expressed by the agent set $\Lambda$. Such cases are not in line with our definition of CRL, so we instead opt to visualize $\Lambda_B$ as a subset of $\Lambda$. Next, notice that the basis is a subset of both the agents that always reach the basis and the agents that sometimes reach the basis—this follows directly from the combination of Proposition C.14 and point (3.) of Theorem 4.3. By similar reasoning from Proposition C.14, we know that the set of agents that always reaches $\Lambda_B$ is a subset of the agents that sometimes reach the basis. Further, since sometimes and never reaches are negations of one another (Remark 3.2), observe that the two sets are disjoint, and together comprise the entirety of $\Lambda$. Lastly, we know that the set of optimal agents, $\Lambda^*$, contains only agents that never reach the basis, and thus the set $\Lambda^*$ is disjoint from $\Lambda_B$ and the set of agents that sometimes reach $\Lambda_B$.

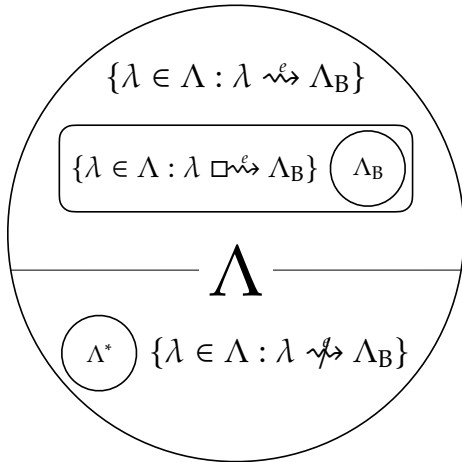

Figure 3: A depiction of the division of a set of agents $\Lambda$ relative to a basis $\Lambda_B$ through the reaches operator in CRL.

