# OpenReview forum: "A Definition of Continual Reinforcement Learning"
_NeurIPS.cc/2023/Conference — NeurIPS 2023 poster_

### Official Review · Reviewer_1ia1 · 2023-06-14

**Soundness:** 3 good
**Presentation:** 3 good
**Contribution:** 3 good
**Rating:** 7
**Confidence:** 3

**Summary:**

The paper looks at developing a foundation for continual reinforcement learning (CRL). The authors develop definitions and insights, aiming to formalize the intuitive concepts in continual learning fields. They also provide two examples of CRL to illustrate the difference between traditional RL and CRL, which is viewing learning as endless adaptation instead of finding a solution.

**Strengths:**

1, The concept of CRL as agent designer aiming to identify an optimal agent among available agents is innovative.

2, The use of rigorous math tools and definition in this paper could provide valuable insight into continual learning and potentially inspire future CRL algorithms development.

**Weaknesses:**

1, Though it's not the focus and goal of this paper, the idea of agent basis contains redundancies and may hinder the transformation of it into a memory and computation efficient CRL algorithm.

2, The training process of agent basis is unclear in Figure 2 (b). I guess two examples are presented to show the effectiveness of CRL, but it's not surprising that CRL can beat RL without adaptation in these changing envs. More online learning benchmarks would help to achieve a fair comparison.

3, The paper is not well structured, and only few related works are mentioned. Continual Learning is discussed in many fields, with different names (e.g., online learning, few-shot adaptation, adaptive control), it would be valuable to provide a broader literature review.

**Questions:**

The high-level idea of CRL training is understandable, but I am unsure about the training details in Section 4.2 CRL examples. Including a pseudocode description could clarify the training process and help understand the implementation details.

**Limitations:**

Overall, the paper requires a lot of additional work to become publishable. Adding figures and pseudocode would enhance readability and understanding. Furthermore, conducting more experiments to demonstrate the effectiveness of the proposed CRL idea would strengthen the paper's contributions.

---

> ### Author Rebuttal · Authors · 2023-08-08
>
> First, we would like to thank the reviewer for reviewing our paper. We address the reviewer’s primary questions and concerns below, and will plan to update the paper in line with the reviewer’s suggestions.
>
>
> > 1, Though it's not the focus and goal of this paper, the idea of agent basis contains redundancies and may hinder the transformation of it into a memory and computation efficient CRL algorithm.
>
> Can you say more here? We do not understand the comment. It is unclear in what sense a basis contains redundancies. We show in Theorem 5.1.4+5.1.5 that it is a necessary property of every CRL problem that the design space of agents contains redundancies in a particular sense (with proofs provided in Appendix C). We suggest that, on a certain view, this might be unsurprising as the space of agents generated by a fixed agent basis will likely contain agents that are similar in their search through the agent basis. Critically, this redundancy need not apply to an agent basis. And, it is unclear to us what is meant by “...may hinder the transformation of it into a memory and computation efficient CRL algorithm”. If the reviewer could clarify we are happy to discuss further.
>
> > 2, The training process of agent basis is unclear in Figure 2 (b). I guess two examples are presented to show the effectiveness of CRL, but it's not surprising that CRL can beat RL without adaptation in these changing envs. More online learning benchmarks would help to achieve a fair comparison.
>
> The examples in Section 4.2 are intended to illustrate that our definition of continual RL can directly accommodate standard instances of continual learning—the first case is an instance of the typical multi-task or lifelong RL setting often used in prior work (see, for instance, Wilson et al. 2007 or Brunskill and Li 2014), and the second case is exactly continual supervised learning as described in the survey by Mai et al. 2022. In the first example, the two agents deployed are both tabular Q-learning, where the only difference is the choice of the annealing schedule for the step-size parameter. In the “$\alpha=C$ case, the step-size is a fixed constant, whereas in the $\alpha=$anneal case, the step-size is annealed to zero in the limit.
>
> > 3, …only few related works are mentioned. Continual Learning is discussed in many fields, with different names (e.g., online learning, few-shot adaptation, adaptive control), it would be valuable to provide a broader literature review.
>
> We are happy to include a broader and deeper discussion of related work on continual learning. We will plan to focus this discussion on the explicit relationship between our definition and past approaches to continual RL from the work by Ring and the recent continual RL survey by Khetarpal et al.. As mentioned in our reply to all reviewers, the work by Ring emphasizes the generality of the environment model and the reward function, while the survey by Khetarpal et al. focuses explicitly on non-stationary MDPs, similar to typical work on multi-task RL. We will include discussion showing the sense in which we accommodate and extend both of these views.
>
> > The high-level idea of CRL training is understandable, but I am unsure about the training details in Section 4.2 CRL examples. Including a pseudocode description could clarify the training process and help understand the implementation details.
>
> Can the reviewer clarify which details would be helpful to hear more about? Both algorithms are simply tabular Q-learning with different step-size annealing schedules. We are more than happy to include pseudocode (space permitting, or in the appendix) if that would help.

---

> > ### Comment · Reviewer_1ia1 · 2023-08-20
> >
> > Thanks the authors for their thorough response and it really helps my understanding.
> >
> > My concern was that the agent needs to switch between the a set of base agents and this set can be large and needs more training power.
> > Like sec 4.2 CRL example, we have four policies to train and then learn a particular agent by searching over the base agents. Please let me know if I get it right.

---

> > > ### Author Response · Authors · 2023-08-20
> > > **Response to Comment**
> > >
> > > We thank the reviewer for following up, and we are glad to hear our response has helped.
> > >
> > > > My concern was that the agent needs to switch between the a set of base agents and this set can be large and needs more training power. Like sec 4.2 CRL example, we have four policies to train and then learn a particular agent by searching over the base agents. Please let me know if I get it right.
> > >
> > > To clarify, when we talk about agents switching between elements of an agent basis, we do so in a general way: we mean that _every agent_, including all of those reinforcement learning algorithms we have designed and implemented to date, can be understood in this way. For instance, consider DQN (Mnih et al. 2015). The Q-network maintained by this algorithm at any point of time can be thought of as the currently active element of the basis. Then, through learning from experience, the algorithm updates the parameters of this Q-network, thereby updating the currently active element of the basis. In this way, all assignments of parameters to the network is the agent basis (the set), and any element of this set is one of the possible elements of the basis (a specific Q-network, and thus, a policy). In the example in section 4.2, there are not four policies to train---both agents (standard Q-learning and Q-learning with a fixed step-size parameter) search through the space of Q functions, roughly.
> > >
> > > As a result, we do not quite understand the point that that an agent needs more training power, as the above perspective applies to all agents.
> > >
> > > We do agree that continual RL problems are likely to be _difficult_ learning problems in general: learning to adapt endlessly is likely _harder_ than learning to find a single solution to a problem in many cases. We believe it is important for the community to embrace and confront this difficulty, which is why we think it is useful to carefully define the problem.
> > >
> > > Does that help? We are happy to help clarify or discuss further.

---

> > > > ### Comment · Reviewer_1ia1 · 2023-08-20
> > > >
> > > > Thanks for your clarification.
> > > >
> > > > Now I get your point: the agent basis is the previous policies. This is interesting and it wouldn't cause extra training power.
> > > > You have solved my concerns, and I will adjust the paper score accordingly.

---

### Official Review · Reviewer_tAG2 · 2023-07-05

**Soundness:** 2 fair
**Presentation:** 2 fair
**Contribution:** 2 fair
**Rating:** 4
**Confidence:** 2

**Summary:**

In this paper, the authors develop a simple mathematical definition of the continual RL problem. These definitions, insights, and results formalize many intuitive concepts at the heart of continual learning, and may open new research pathways surrounding continual learning agents.

**Strengths:**

The mathematical definition of continual reinforcement learning give us some inspirations when thinking of the continual learning, multi-task RL, continual supervised learning, and so on.

**Weaknesses:**

1.The abstract is too simple that readers cannot get enough information and key ideas from it.
2.This paper is not easy to follow, for the overall mathematical definitions and analysis. I hope the authors can give us more cases or experiments using the definitions proposed in this paper to highlights the benefits of following such definitions.

**Questions:**

1.What information can we get from Figure 2(a) Switching MDP Visual?  Do authors just want to emphasis “Each underlying MDP shares the same state space and action space, but varies in transition and reward functions”?
2.How can we prove or evaluate the definitions, the proposed operators, and so on?
3.Can authors add some ablation studies in this paper?

**Limitations:**

Please modify the paper as the weaknesses and questions above, and add limitation analysis in the paper.

---

> ### Author Rebuttal · Authors · 2023-08-08
>
> First, we would like to thank the reviewer for reviewing our paper. We address the reviewer’s primary questions and concerns below, and will plan to update the paper in line with the reviewer’s suggestions.
>
> > 1.The abstract is too simple that readers cannot get enough information and key ideas from it.
>
> We had originally thought a simple abstract would be a good choice, but in hindsight, we agree with the reviewer and will modify it accordingly. We will plan to change our abstract to the one mentioned in the rebuttal reply to all reviewers, and are open to suggestions for improvement.
>
> > 1.What information can we get from Figure 2(a) Switching MDP Visual? Do authors just want to emphasis “Each underlying MDP shares the same state space and action space, but varies in transition and reward functions”?
>
> Yes, that is correct; we would like to give the reader a clear sense of the learning problem by emphasizing that each individual MDP shares a state-action space, but differs in its other components.
>
> > 2.How can we prove or evaluate the definitions, the proposed operators, and so on?
>
> We believe that our definitions and operators should (1) capture intuitions and easily accommodate known cases of continual learning, (2) hold up under scrutiny, and (3) provide a language for carefully discussing important phenomena related to continual learning. We believe that our definition and the operators accomplish these three feats.
>
> For instance, our CRL definition formalizes the intuition that “the best agents keep learning forever”, and the two examples we provide in Section 4.2 showcase that our definition can accommodate known special cases. Moreover, the necessary properties we establish of the CRL definition and the operators (Theorem 5.1, Theorem 5.2, Theorem 5.3) provide initial character that goes beyond the contents of the definitions themselves. For example, in defining the generates operator, it may not be obvious that deciding whether a given basis generates an agent set in an environment is undecidable–Theorem 5.2 proves this is in fact the case. In this sense, we believe our definition and operators satisfy the three above criteria (capture intuitions and accommodates known cases, holds up under scrutiny, and provides a language for discussing phenomena related to continual learning).
>
> > 3. Can authors add some ablation studies in this paper?
>
> It is unclear to us what exactly an ablation would be in this case, and to what end we need one. Can you please elaborate?

---

### Official Review · Reviewer_DFWz · 2023-07-06

**Soundness:** 2 fair
**Presentation:** 3 good
**Contribution:** 2 fair
**Rating:** 4
**Confidence:** 3

**Summary:**

The paper proposes a mathematical formulation for the problem of continual reinforcement learning in an infinite horizon setting.

**Strengths:**

1. The authors propose a new mathematical formalism for continual RL.
2. The paper can be of interest to mathematically inclined readers and could potentially lead to theoretical results in continual reinforcement learning.


**Weaknesses:**

1. Abstract: there already are foundations for CRL. You cited a survey paper [16] and multiple other existing works. As unconventional as it is, I wouldn’t be against a short abstract, but I do not believe the current one will do.
2. Missed important opportunities to connect your formalism with existing (continual) RL. E.g. eq 2.1 states an infinite horizon, which is required for example in Theorem 3.1, but the very standard “infinite horizon” vocabulary never appears. I think the paper is not connected enough to existing (continual) RL literature.
3. Generally, I’m not convinced by motivations. The paper says it is important, but I did not really see a convincing motivation for abstracting the problem of continual RL. The theorems provide results that can be conveyed in significantly easier ways with existing formulations: e.g. non commutativity in thm 5.2 signifies that in continual RL, some policy transitions cannot be “unlearnt” in some settings. The formalism incurs a very significant overhead over such high level language – I believe that the authors can improve the motivations to justify why their formalism is worth it.


**Typos and suggestions:**
1. L10: we instead of We
2. L41: I would use $\mathcal{Z}$ instead of $\mathcal{X}$ so it flows better in the following sentence.
3. L84: phrasing “instead opt for simply the bounded…”
4. L93: did you mean the sum of rewards to start at index 0 ? or t+1, t+2, etc.
5. L95: I know it got absorbed into v, but I would still prepend this line with “Given <initial conditions> $h_0$, any tuple…” since your work aims to formalise rigorously the problem.
6. “We use as if in the sense of the positive economists” this is terminology that should be defined in the paper.
7. L334: “generates is commutative” -> are
8. L338: I recommend not defining objects used in a main body theorem in the appendix.
9. I believe that the insights, as stated in the conclusion, are trivial; perhaps “insight” isn’t the word that should be used (using the Cambridge dictionary definition of “insight” “ a clear, deep, and sometimes sudden understanding of a complicated problem or situation” – I do not believe the idea for example that agents in CRL either can or cannot find a set of policies/behaviours is insightful).


**Questions:**

No.

**Limitations:**

Yes.

---

> ### Author Rebuttal · Authors · 2023-08-08
>
> First, we would like to thank the reviewer for their time and energy in reading and reviewing our paper. We address the reviewer’s primary questions and concerns below, and will plan to update the paper in line with the reviewer’s suggestions.
>
> > Abstract: there already are foundations for CRL. You cited a survey paper [16] and multiple other existing works. As unconventional as it is, I wouldn’t be against a short abstract, but I do not believe the current one will do.
>
> This is a great point, and we are happy to extend our abstract in light of the feedback. Our proposed abstract draft is provided in the reply to all reviewers ("[Overall Response]")
>
> > Missed important opportunities to connect your formalism with existing (continual) RL. E.g. eq 2.1 states an infinite horizon, which is required for example in Theorem 3.1, but the very standard “infinite horizon” vocabulary never appears. I think the paper is not connected enough to existing (continual) RL literature.
>
> We agree, and recognize the importance of discussing the relationship between our formalism and prior approaches to continual RL. We will include a discussion that elucidates these relationships further, with a focus on the thesis by Ring and the recent survey by Khetarpal et al. Further details of this relationship are mentioned in the "[Overall Response]" to all reviewers.
>
> > Generally, I’m not convinced by motivations. The paper says it is important, but I did not really see a convincing motivation for abstracting the problem of continual RL... I believe that the authors can improve the motivations to justify why their formalism is worth it.
>
> We believe that precise definitions are essential for clarity in science. For example, prior to the discovery of pseudo-randomness, much work in theoretical computer science relied on ad-hoc heuristics for producing random sequences to the detriment of the effectiveness of these approaches. Then, a series of papers carefully defined what it would mean for a sequence to be pseudo-random (for instance, in “The definition of random sequences” by P. Martin-Lof, and “Theory and applications of trapdoor functions” by Yao). On Yao’s view, a pseudo-random sequence is one that a bounded adversary cannot distinguish from truly random. Achieving this clarity allowed for pseudo-random number generators to be developed, and for the community to direct its research at the right concepts, and toward the right objectives. In our view, continual reinforcement learning is at a similar stage in its scientific development to pseudo-randomness in its infancy: we lack precise definitions for what, exactly, it is we are conceptualizing. Some people think of non-stationarity, others think of computational constraints, while others think of multi-task learning and transfer. Our goal in developing a single abstract definition is to encompass all of these views as special cases, and the point of formalizing it is to remove any ambiguity in the use of terms like “continual learning”, or “continual learning agent”. We can include more of this motivation in the paper, and are happy to discuss further.
>
> > I believe that the insights, as stated in the conclusion, are trivial; perhaps “insight” isn’t the word that should be used… I do not believe the idea for example that agents in CRL either can or cannot find a set of policies/behaviours is insightful.
>
> Thanks for the feedback here, and we are happy to consider alternative words to replace “insight”. However, we would like to clarify one point, and justify another. First, to clarify: the second insight does not state an agent in CRL can or cannot find a set of policies. This does not reflect the semantics of Remark 3.2. The remark says that _every_ agent can be classified into exactly one of two families: (1) the agent eventually stops its search over its corresponding basis, or (2) searches forever. This is regardless of the learning setting, environment, or reward function. The generality and precision of the statement is why we believe it is insightful. Second, to justify further why we chose the term “insight”: we provide a formalism of an intuitive fact that holds true of every possible agent in a way that, in hindsight, the fact seems clear and even obvious. Even if the intuition is obvious (in hindsight), knowing how precisely to state this fact rigorously in a way that holds true of every possible agent is non-trivial (in our opinion). In this sense, we take the precise formalisation of the intuition to be insightful. Moreover, the first insight (conveyed through Theorem 3.2), we take to be the stronger of the two. We are happy to discuss further, and if the reviewer is unswayed by the above, we are happy to change the term “insight” to something weaker.
>
> > L95: I know it got absorbed into v, but I would still prepend this line with “Given <initial conditions> ℎ0, any tuple…” since your work aims to formalise rigorously the problem.
>
> Good point, fixed!
>
> > “We use as if in the sense of the positive economists” this is terminology that should be defined in the paper.
>
> We are happy to include an expanded description of this terminology in the paper.
>
> > L338: I recommend not defining objects used in a main body theorem in the appendix.
>
> This is a fair point. We will move the definition of “always reaches” to the main body of the paper.
>
> > Other Typos and Writing suggestions:
>
> Thanks! We have fixed the remaining suggested typos and writing suggestions.

---

> > ### Comment · Reviewer_DFWz · 2023-08-14
> >
> > I would like to thank the authors for the extensive replies to every reviewer and acknowledging the importance of rethinking the abstract and litterature/contextualising.
> >
> > > We believe that precise definitions are essential for clarity in science. For example, prior to the discovery of pseudo-randomness, much work in theoretical computer science relied on ad-hoc heuristics for producing random sequences to the detriment of the effectiveness of these approaches. Then, a series of papers carefully defined what it would mean for a sequence to be pseudo-random (for instance, in “The definition of random sequences” by P. Martin-Lof, and “Theory and applications of trapdoor functions” by Yao). On Yao’s view, a pseudo-random sequence is one that a bounded adversary cannot distinguish from truly random. Achieving this clarity allowed for pseudo-random number generators to be developed, and for the community to direct its research at the right concepts, and toward the right objectives. In our view, continual reinforcement learning is at a similar stage in its scientific development to pseudo-randomness in its infancy: we lack precise definitions for what, exactly, it is we are conceptualizing.
> >
> > I agree, as someone who often has to be on the other end of this argument. However, in my opinion, the less accessible a formalism is (and I do believe the one presented here is on the more abstract end of the spectrum), the more burden there is to motivate it. For example, Balduzzi et al.'s *The Mechanics of n-Player Differentiable Games* was a highly impactful paper not only due to its simplicity, but also for proposing in the paper new analyses using their formalism. I understand that the problem here may warrant a more complex formalism, but I still believe proposing something using that formalism is important.
> >
> > > Some people think of non-stationarity, others think of computational constraints, while others think of multi-task learning and transfer. Our goal in developing a single abstract definition is to encompass all of these views as special cases, and the point of formalizing it is to remove any ambiguity in the use of terms like “continual learning”, or “continual learning agent”.
> >
> > That's a unrelated to your submission but 1) I believe *non-stationarity* already captures multi-task learning and transfer, and 2) are those terms really ambiguous ? I have not experienced much disagreement over the definitions of CL that has somehow impeded discussions/led to misunderstandings.
> >
> > > First, to clarify: the second insight does not state an agent in CRL can or cannot find a set of policies. This does not reflect the semantics of Remark 3.2. The remark says that every agent can be classified into exactly one of two families: (1) the agent eventually stops its search over its corresponding basis, or (2) searches forever.
> >
> > I might be missing something re: a third possible outcome. The way I'm reading this is "either A or not A" where A = "the agent eventually stops its search". Again, I assume I'm missing something, so could the authors clarify this point further ? Also, I believe what I stated is the loose equivalent of that, under equating agent with policy, and assuming the search stopping = "finding", but I may very well be wrong.
> >
> > > Second, to justify further why we chose the term “insight”: we provide a formalism of an intuitive fact that holds true of every possible agent in a way that, in hindsight, the fact seems clear and even obvious. Even if the intuition is obvious (in hindsight)
> >
> > While I understand the point of having it stem from the formalism, and you might argue the following is semantics, but it's more foresight than hindsight. The formalism is developed (just like in natural sciences) with the goal of matching current knowledge of CL, but capturing them with a more rigorous formulation. After all, that is the motivation of the paper.

---

> > > ### Author Response · Authors · 2023-08-18
> > > **First Response**
> > >
> > > We thank the reviewer for their thoughtful reply. We have two immediate reactions:
> > >
> > > > I might be missing something re: a third possible outcome. The way I'm reading this is "either A or not A" where A = "the agent eventually stops its search".
> > >
> > > We now believe we agree here: the remark is pointing out a claim of the form "either A or not A", exactly as you suggest, and A = "the agent eventually stops its search", as you suggest. Our original comment was indicating that we believed there was misunderstanding on what A was ("finding a set of policies" vs. "agent stopping its search"). We believe that we are all now on the same page.
> > >
> > > And, regarding the the choice of the term "insight":
> > >
> > > > While I understand the point of having it stem from the formalism, and you might argue the following is semantics, but it's more foresight than hindsight. The formalism is developed (just like in natural sciences) with the goal of matching current knowledge of CL, but capturing them with a more rigorous formulation. After all, that is the motivation of the paper.
> > >
> > > Thanks, that resonates. Our main points here are: that: (1) formalizing intuitions about _all_ agents into rigorous statements that reflect common knowledge and intuitions is non-trivial, and (2) the first "insight" (Theorem 3.1) is really intended to be the more "insightful" of the two.
> > >
> > > We want to emphasize that we are more than happy to change the paper to move away from the term "insight", as we appreciate and understand where the reviewer is coming from with this point.

---

### Official Review · Reviewer_EKup · 2023-07-19

**Soundness:** 4 excellent
**Presentation:** 4 excellent
**Contribution:** 3 good
**Rating:** 7
**Confidence:** 4

**Summary:**

- This paper lays out a foundation for continual reinforcement learning (CLR) from the ground up—establishing definitions for the purpose of building towards a technical definition of CRL itself, proving various properties of CRL, and through employing simple CRL examples, demonstrating some of these properties.

**Strengths:**

- The writing is exceptionally clear and precise.
- The framework is well thought-out; the definitions are clear and useful, and the framework as a whole is both technical and intuitive. Importantly the provided definition of CRL seems to get at the root of what researchers have always vaguely meant by CRL.
- There is a rich foundation set for future work (e.g. exploring “connections between our formalism of continual learning and some of the phenomena at the heart of recent empirical continual learning studies, such as plasticity loss, in-context learning, and catastrophic forgetting”; line 364).

**Weaknesses:**

- I understand the appeal of a short abstract (i.e. it is to the point, and (albeit somewhat pompously) implies some special finality or importance to the contributions in the work), but the resolution of the abstract, for good reason, usually sits between the resolution of the title and the paper itself. The abstract in this case does not provide much more information than the title, so any reader that wants a quick overview of the paper must read the paper itself, defeating the purpose of the abstract. I suggest the authors to reconsider employing such a short abstract.
- I feel that the paper could have developed more connections to previous literature, performed more experiments building off of this framework, etc., i.e. more validation for why we should care about this framework. Better (nonstationary) environments to evaluate agents on? Agent structures? Assessment of how well current methods fit into/exploit elements of this framework? The framework is nice, but it would be useful to see how it can situated into current literature, either by retroactively analyzing current agents/environments w.r.t. this framework, or designing new agents/environments from the principles this framework provides.
- Maybe more real-world examples of changing environments? Or where we want the behavior to change in the same environment (but then we just have an oscillating agent…). Something about the environment changing (or being better understood over time) is fundamental to continual RL)
- It took me a number of readings to understand agent bases; i.e. the relationship between $\Lambda$ and $\Lambda_B$.

**Questions:**

- Do we really want agents that will adapt their policy forever? Or are the environments we care about simply non-stationary, i.e. because the world changes, and the agent is required to learn more about the environment in order to continue to act optimally? Maybe the policy (in some high-level sense) should remain the same, and it is the world model that should be adapted? E.g. suppose the expressed behavior of the agent policy can be boiled down to “helping humans”, and this policy is fixed, given the agents current understanding of the world as context. Perhaps only this context needs to be updated, i.e. as the world changes, and/or the agent learns more about the existing world. This seems to fit within the proposed framework, if we consider the inferred world model as part of the agent. And while it does seem likely that the discovery of new contexts would require updating the policy, it still appears to me to be beneficial to disentangle these aspects explicitly. What are the authors' thoughts on this?


**Limitations:**

- The authors have discussed the limitations thoroughly, and I have listed them above as a strength as there are many directions ripe for future work.

---

> ### Author Rebuttal · Authors · 2023-08-08
>
> First, we would like to thank the reviewer for their time and energy in reading and reviewing our paper. We address the reviewer’s primary questions and concerns below, and will plan to update the paper in line with the reviewer’s suggestions.
>
> > The abstract in this case does not provide much more information than the title, so any reader that wants a quick overview of the paper must read the paper itself, defeating the purpose of the abstract. I suggest the authors to reconsider employing such a short abstract.
>
> We had originally thought a simple abstract would be a good choice, but we hear the reviewers' suggestion, and will modify it accordingly. We will plan to change our abstract to the one listed in the rebuttal comment provided to all reviewers (see "[Overall Response]" above). We are happy to make any further changes based on your feedback.
>
> > I feel that the paper could have developed more connections to previous literature,
>
> This is a valid point, thanks! We have two reactions:
>
> First, in light of this suggestion, we will include additional text discussing explicit connections with previous approaches to continual reinforcement learning, with an emphasis on the work by Ring and the recent continual RL survey by Khetarpal et al. Succinctly, the work by Ring emphasizes the generality of the environment model and the reward function (which we also adopt), while the survey by Khetarpal et al. focuses explicitly on non-stationary MDPs, similar to typical work on multi-task RL (as in work by Brunskill and Li, for instance). Our definition directly accommodates and extends both of these views, and we are happy to include discussion on this topic in the paper.
>
> Second, one of the primary goals of this paper is to promote a change of mindset—popular thinking in RL tends to view the learning problem as the search for one policy (typically, the optimal policy of an MDP). However, in some cases it is better to think of learning as endless adaptation. We suspect this change in perspective will be increasingly important as agents are deployed in the real world, which is notoriously messy and changes regularly. While this shift in mindset is not new, we do believe we offer a concrete way to conceptualize of this change in perspective that can help promote new RL research aligned with the problem facing agents we are building today.
>
> > …performed more experiments building off of this framework, etc., i.e. more validation for why we should care about this framework. Better (nonstationary) environments to evaluate agents on? Agent structures?
>
> Regarding additional validation, evaluation, and environments: we agree with the spirit of this point, and intend to carry out further work motivated by the definition. However, we do believe that the essence of this paper is about carefully defining the problem, and that further evaluation of the kind suggested is best deferred for follow-up work to give it the care and space it deserves.
>
> > Q: Do we really want agents that will adapt their policy forever? Or are the environments we care about simply non-stationary, i.e. because the world changes, and the agent is required to learn more about the environment in order to continue to act optimally?
>
> This is another great point, and one we explored in depth as part of this work. We believe it is an open question as to whether the two concepts (policy adaptation vs. learning more about the environment) are different, deeply connected, or the actually same concept just viewed from different lenses. Roughly, we can see that an agent that must learn more about its environment to act optimally but does not ever adapt its policy is a peculiar case, and one that we suggest should not be viewed as continual learning (that is, when the agent can be optimal while _never_ updating its policy). We suggest there is a deep connection between the two views, and are actively exploring this connection in follow up work. We are happy to comment on their connections in the discussion section of the paper, and to discuss this further.

---

> > ### Comment · Reviewer_EKup · 2023-08-16
> >
> > I thank the authors for their thorough response. I maintain my positive opinion of this paper, especially in light of the authors' (tentative) updates to the draft. I believe this work is ambitious and useful to the community; and that the potential for this work to spur more technically-grounded future research in this area far outweighs the weaknesses pointed out by myself and other reviewers---weaknesses I also believe have been adequately addressed by the authors.

---

### Official Review · Reviewer_oHpE · 2023-07-23

**Soundness:** 3 good
**Presentation:** 3 good
**Contribution:** 3 good
**Rating:** 7
**Confidence:** 2

**Summary:**

The authors propose a new definition of continual reinforcement learning where an optimal continual learning agent will not converge to a fixed policy. This is formalized through the generate operator, which defines the "searching" between different policies; and the reach operator, which defines whether the agent will stop searching.

**Strengths:**

1. This paper provides a unique definition of the continual learning problem, which is formalized by their mathematical framework.
2. Based on their framework, the authors were able to prove a number of interesting theorems.

**Weaknesses:**

1. It is not immediately obvious how this definition can inspire new learning algorithms.
2. The underlying concept that an optimal learning agent should not converge is not surprising. The author also mentioned that it's an "unsurprising conclusion that it is better to track than converge" for their toy non-stationary environment.

**Questions:**

1. Can you elaborate on how this formal definition of continual learning can address practical problems such as plasticity loss, in-context learning, and catastrophic learning as mentioned in the Discussion section?

**Limitations:**

The authors mention the limitations implicitly as future work in the discussion section. Essentially, since this paper is almost purely theoretical, they treat the lack of empirical study as future work or limitation.

---

> ### Author Rebuttal · Authors · 2023-08-08
>
> First, we would like to thank the reviewer for their time and energy in reading and reviewing our paper. Below, we respond to each in point detail, and we are happy to update the paper to reflect the reviewer’s suggestions, and to continue the discussion.
>
> > It is not immediately obvious how this definition can inspire new learning algorithms.
>
> This is a valid point. Our view is that this paper is about clarifying what problem we are actually studying. Doing so will remove the ambiguity that has surrounded continual learning, and allow us as a community to ensure we are studying the right problem. Further, this clarity provides new language to talk about agents and continual learning more carefully. The promise, then, is that exploration of these new tools can yield new perspectives on the design of algorithms. For instance, our work emphasizes the view that an agent can be understood as a choice of a learning rule (definition 3.2) and a choice of agent basis (definition 3.1). These two objects provide a new abstraction on the components of an agent. Furthermore, this abstraction may yield new kinds of learning rules that can inform algorithm design. As an example, we might consider learning rules that are guaranteed to produce a continual learning agent of the kind defined by definition 4.1—what are the necessary and sufficient conditions of such a learning rule? Answering this question will give a path toward defining algorithms that are guaranteed to produce continual learning agents (which, as Dohare et al. showed recently, is not true of many standard learning rules). Or, as we explore briefly in Appendix C, we might consider learning rules that capture certain intuitive families of agents, such as a model-based learning rule. Combined with tools for thinking about continual learning, we could imagine delineating between continual model-based agents and regular model-based agents; such a distinction could give rise to new algorithmic principles and other algorithmic primitives, but working out the precise details is beyond the scope of the present submission.
>
> > The underlying concept that an optimal learning agent should not converge is not surprising.
>
> We agree, and we take the intuitive appeal of the statement “an optimal learning agent should not converge”, as a positive characteristic of our definition. Our goal is to make this intuition mathematically precise, which we believe we have done.
>
>
> > Q:  Can you elaborate on how this formal definition of continual learning can address practical problems such as plasticity loss, in-context learning, and catastrophic learning as mentioned in the Discussion section?
>
> This is a great question, and our reasoning will be similar to our response to the first point above. Now that we have a precise mathematical language for talking about continual learning agents, we believe we are well positioned to unpack and examine the kinds of phenomena that regularly arise in continual learning. We provide a similar answer in the response to all reviewers ("[Overall Response]"), but for simplicity, here are our thoughts. As an example, let’s consider plasticity loss. We suggest that plasticity loss can be modeled precisely using the conceptual tools we develop: _plasticity_ is the fraction of an agent’s corresponding agent basis that remains reachable to the agent over time, and plasticity _loss_ is any reduction of this number. We foresee pathways to both diagnose agents from a new perspective, and to design algorithms based around this measure; for example, by developing learning rules (def 3.2) that provably maintain plasticity. This unlocks a new point of emphasis from designing learning algorithms that is grounded in a rich mathematical toolkit. Or, we can understand in-context learning precisely in terms of the kinds of agent bases that agents search through—certain kinds of bases might be sufficiently rich to be capable of in-context learning, while others are not. Therefore, in the same way that we precisely define a continual learning agent in definition 4.1, we might unlock a precise definition of an in-context learning agent, at which point we can study the necessary and sufficient conditions required of such an agent. This is one example, but the same kind of analysis is possible for a variety of agent-centric properties such as plasticity loss and catastrophic forgetting.

---

> > ### Comment · Reviewer_oHpE · 2023-08-17
> >
> > I'd first like to thank the authors for the detailed response.
> >
> > I think this is indeed an interesting direction, and I raised my score accordingly.
> >
> > Perhaps a bit outside the scope of this work, but I feel that actually analyzing plasticity using a toy task under the proposed definition, for example, can make the paper a lot more compelling.

---

### Official Review · Reviewer_9sbq · 2023-07-24

**Soundness:** 3 good
**Presentation:** 4 excellent
**Contribution:** 3 good
**Rating:** 7
**Confidence:** 3

**Summary:**

This is an ambitious paper.  The paper notes that the problem of "Continual Reinforcement Learning" lacks a rigorous definition and seeks to provide one.  It mathematically defines the reinforcement learning problem and then explores the conditions in which an instance of the RL problem is a CRL problem.

The legwork is done through the new "generates" and "reaches" operators.

The "generates" operator accepts two arguments: 1. a set of "basis" agents, and 2. a particular agent (which may not be in that set, but is exhibits some combined behavior of the basis agents).  It returns boolean truth value.  The "generates" operator expresses if there exists a learning rule that can search over "basis" agents (with respect to an environment / historical observations) and find a combination of them that is equivalent to the particular agent in the argument. This function is also undecidable in general.

Loosely, the "reaches" operator (wrt an environment) accepts two arguments: 1. a particular agent 2. a set of "basis" agents It returns a boolean truth value.  It returns True when an agent's behavior settles on an element in the basis set.  The authors actually define a more precise "sometimes reaches" and "never reaches" operator. This is also undecidable.

A learned agent will always choose a behavior equivalent to some basis agent in the context of the environment and history.

Using these formalisms, a CRL agent is one that never reaches an agent basis, and a CRL problem is one where the best agent is a CRL agent.

Notably, a problem is a CRL problem depending on the choice of basis.

The paper performs mathematical analysis on these operators, relates them to two example instances of CRL, and provides a thoughtful discussion for new questions that can be asked about agent basis sets.


**Strengths:**

The informal definition "An RL problem is an instance of CRL is the best agents never stop learning" is intuitive and clear.

I always appreciate explicit definitions for notation. (although defining logical negation, and the quantifiers might not be necessary in a conference paper). I type checked several of the equations and everything seems to work out nicely. The theory gives a good feel for how someone might implement a framework for exploring agents in the language of this theory using a strongly typed language.

The paper gives explicit examples and experiments demonstrating intuitive properties of this theory in the setting of Q-learning and finite Markov decision processes. As well as a more complex example in continual supervised learning.

The paper proves several basic properties of the new reaches and generates operators. I have not checked the proofs for correctness, but I also don't see any obvious problems.

Section 5 made me delete a lot of my questions because it answered them. The idea of exploring choices of the basis is mathematically interesting.

The paper raises more questions than it answers.


**Weaknesses:**

The abstract is terse - problematically so. Even if this succeeds at being a theoretically sound definition of continual reinforcement learning, I think it is important to elaborate at least a little bit more. Note this is the main reason I'm rating this a 5 / 10. Given a better abstract I think this is a 7/10, but I think it is critical to give a better overview to the researchers' attention you are competing for.

A 9 page paper with a 20 page appendix full of proofs may be better suited for a journal.

The paper raises more questions than it answers.

**Questions:**

On line 47 the authors explicitly call out that the set A and O are countable.  Does this imply that some other (non-numeric - i.e. non-ℝ) sets may be uncountable in this definition?

I think the order of appendix A and appendix B should be switched. Provide the notation first. (The table was very helpful in reviewing).


I became confused when I first read Theorem 3.1. Particularly when the selected agent was not an element of the agent basis. In hindsight it makes sense that the new agent is a (linear?) combination behaviors in the agent basis set, but I think the paper could make that more clear. It wasn't intuitive what "switches" meant to me. Or that lambda(h) in Figure 1 was a combination of the basis agents conditioned on history. (At first I thought the learning rule was just choosing an element of the basis set).

In Theorem 3.1 are the infinite choices of basis countable?

If a problem's status as CRL or not depends on the choice of basis, and there are an infinite choices for the basis, then is this framework all that useful for thinking about problems?

Theorem 5.1.2 suggests that it is possible to reduce any CRL problem to an RL problem via a change of basis.

Is it possible to reduce any CRL problem to a RL problem by finding a function that changes the basis? Are there problems where such a mapping does not exist?

Have you formalized any of the proofs into a proof assistant like Lean? I took a peek at the appendix, but I simply don't have the time to verify these proofs. Being able to point to an automatically checked proof in each theorem would greatly increase the quality of the paper.

**Limitations:**

An optimal CRL agent could be difficult to rectify with the alignment problem. In contrast, it might be the case that deciding if a CRL agent will remain aligned with its creators is undecidable. That question is worth exploring.

---

> ### Author Rebuttal · Authors · 2023-08-08
>
> First, we would like to thank the reviewer for their time and energy in reading and reviewing our paper. We recognize the reviewer spent a lot of time understanding and commenting on our work, and we appreciate it. Below, we respond to each in point detail, and we are happy to update the paper to reflect the reviewer’s suggestions, and to continue the discussion.
>
> > The abstract is terse - problematically so…
>
> We had originally thought a simple abstract would be a good choice, but in hindsight, we completely agree with the reviewer, and will modify our abstract accordingly. We will plan to change our abstract to the text provided in the reply to all reviewers ("[Overall Response]"), and are open to suggestions for improvement.
>
> > Swap Appendix A and B
>
> This is a great idea. We will swap the order of these two appendices.
>
> > I became confused when I first read Theorem 3.1. Particularly when the selected agent was not an element of the agent basis..
>
> This is helpful feedback. We can spend time and improve the exposition of this result.
>
> > In Theorem 3.1 are the infinite choices of basis countable?
>
> Good question! The proof strategy of the result involves constructing a countable sequence of basis sets (where each of the sets will generate the original agent in the environment). Thus, the proof strategy only *requires* countably infinitely many bases. However, we suspect that there are augmentations to the proof strategy that also make use of uncountably infinitely many bases, so the argument can likely go through in both cases.
>
> > If a problem's status as CRL or not depends on the choice of basis, and there are an infinite choices for the basis, then is this framework all that useful for thinking about problems?
>
> This is a critical point, and one we have focused on in developing this work. We believe that this point is a feature of the definition, rather than a bug. To see why, we will plan to include the following text in Section 5 following the statement of Theorem 5.
>
> _It is reasonable to ask if the fact that our definition of CRL is basis-dependant renders it vacuous. We argue that this is not the case for two reasons. First, we conjecture that any definition of continual learning that involves concepts like “learning” and “convergence” will have to sit on top of some reference object whose choice is arbitrary. Second, and more important, even though the mathematical construction allows for an easy change of basis, in practice the choice of basis is constrained by considerations like the availability of computational resources. It is often the case that the domain or problem of interest provides obvious choices of bases, or imposes constraints that force us as designers to restrict attention to a space of plausible bases. For example, as discussed earlier, a choice of neural network architecture might comprise a basis—any assignment of weights is an element of the basis, and the learning rule $\sigma$ is a mechanism for updating the active element of the basis (the parameters) in light of experience. In this case, the number of parameters of the network is constrained by what we can actually build. Further, we can think of the learning rule $\sigma$ as something like stochastic gradient descent, rather than a rule that can search through the basis in an unconstrained way. In this sense, the basis is not arbitrary, nor is the learning rule. We as designers choose a class of functions to act as the relevant representations of a mapping from observations to actions, often limited by resource constraints on memory or compute. Then, we use specific learning rules that have been carefully designed to react to experience in a desirable way—for instance, stochastic gradient descent updates the current choice of basis in the direction that would most improve performance. For these reasons, the choice of basis is not arbitrary, but instead reflects the ingredients involved in the design of agents as well as the constraints necessarily imposed by the environment._
>
> We hope the above answers your question, but we are happy to discuss this point further.
>
> > Is it possible to reduce any CRL problem to a RL problem by finding a function that changes the basis? Are there problems where such a mapping does not exist?
>
> This is a really interesting question, and we suspect that crafting a well-formed answer is a great direction to explore further. We do know that, by Theorem 5.1.2, if we were to include all of the optimal agents in the agent basis in question, then the problem is no longer CRL (since there is now an optimal agent that reaches the basis). However, finding a function to construct the optimal agents in this setting is not necessarily feasible. There might exist problems where, depending on the constraints imposed on the basis or set of learning rules, finding this function is either impossible or feasible. Clarifying such settings would be an interesting direction to pursue further.
>
> > An optimal CRL agent could be difficult to rectify with the alignment problem. In contrast, it might be the case that deciding if a CRL agent will remain aligned with its creators is undecidable. That question is worth exploring.
>
> In our view, most agents we as a community build and deploy will be facing a CRL problem, rather than a more traditional RL problem (i.e. those with a fixed solution). As a result, we believe it is an important open question how to frame safety and alignment research around CRL—we take this to be an important line of future work. In our view, it is a positive aspect of the work that it opens new lines of research (that are perhaps more well calibrated to the actual problems facing agents we design) surrounding safety.
>
> > Have you formalized any of the proofs into a proof assistant like Lean?
>
> We have not formalised the proofs into a proof assistant.

---

> > ### Comment · Reviewer_9sbq · 2023-08-17
> >
> > Overall I like this paper, and I think the indended revisions are sufficient for me to raise my rating.
> >
> > > First, we conjecture that any definition of continual learning that involves concepts like “learning” and “convergence” will have to sit on top of some reference object whose choice is arbitrary.
> >
> > I buy this. I'd be interested in seeing a formalization of this conjecture.

---

### Official Review · Reviewer_z8Jz · 2023-07-26

**Soundness:** 3 good
**Presentation:** 3 good
**Contribution:** 3 good
**Rating:** 7
**Confidence:** 3

**Summary:**

In the paper the authors propose a formal framework for reasoning about continual reinforcement learning problems. To this end they introduce mathematical definitions for environments and agents which serve as a basis for defining the general reinforcement learning setting as well as the continual setting. In particular, the authors tie the problem of continual learning to the learning progress an agent makes and two operators describing its limitting learning behaviour. Finally, they provide two examples and derive sevaral characteristics of the proposed operators.

**Strengths:**

The problem of continual RL is most certainly relevant to the community and will arguably become more prominent as real-world agents adapting throughout their deployment become more common-place. To my knowledge the proposed formalism is original in its attempt to characterise this setting formally.

Overall I found the framework sound and its description easy to follow. I agree with the authors that such a formalism has the potential to open up new perspectives on and analysis of (long-term) learning behaviour of agents.

**Weaknesses:**

Throughout the paper the terms "agent" and "behaviour" are used almost interchangeably. However, a formal definition of behaviours is missing, making it somewhat unclear whether or not the two terms are actually meant to be equivalent. If so, the statement "we can understand every agent as implicitly searching over a set of behaviors" needs clarification. If not, I would encourage the authors to formally define what a "behaviour" constitutes.

A related issue is the separation of agents and their learning rules. Arguably, an agent parameterised by a recurrent network is a continual learner as it changes its "behaviour" (in the sense of the one-step policy) based on the history giving rise to its internal state. However, under the provided definition such an agent would not be considered a continual learner unless it updates its weights.

Finally, the paper would benefit from a more in-depth discussion of how previous approaches to continual RL compare to the proposed framework or can be embedded into it. Similarly, the authors mention several avenues for future research such as the examination of catastrophic forgetting - it might be interesting to include at least a characterisation of these problems in the new framework to inspire future work.

**Questions:**

-

**Limitations:**

-

---

> ### Author Rebuttal · Authors · 2023-08-08
>
> First, we would like to thank the reviewer for their time and energy in reading and reviewing our paper. We address the reviewer’s primary questions and concerns below, and will plan to update the paper in line with the reviewer’s suggestions.
>
> > Definition of agent vs. behaviour…
>
> This is a great point, and one we will be sure to attend to carefully in the paper. We use the term “behaviour” early on in (and throughout) the paper to appeal to intuition before we have defined “agent” more precisely in Definition 2.4. The reason for this choice is that we find the phrase, “we can understand every agent as implicitly searching over a set of behaviors” easier to grasp than “we can understand every agent as implicitly searching over another set of agents”. In light of your comment, we will revise the paper to remove references to “behavior” entirely, or make it clear in some cases when we are using the term in a colloquial way to appeal to intuition (and that it will be replaced by the more precise “agent” after Def. 2.4). We hope this helps.
>
> > A related issue is the separation of agents and their learning rules. Arguably, an agent parameterised by a recurrent network is a continual learner as it changes its "behaviour" (in the sense of the one-step policy) based on the history giving rise to its internal state. However, under the provided definition such an agent would not be considered a continual learner unless it updates its weights.
>
> The reviewer makes an excellent point, and one that was at the heart of a lot of the puzzles we thought about as part of this work. In short, as per Theorem 5.1.2, depending on the basis, agents can be understood as learning or not. In an MDP, for instance, do we want to consider a fixed stationary policy as learning, simply because it reacts to a change in the MDP state? Or should we only consider explicit _updates_ to the policy as learning (or something else)? This tension is at the heart of our view of continual learning, and one we fully embrace by allowing the choice of basis to characterize what it means for an agent to learn: an agent is a continual learner with respect to a basis if the agent keeps switching between basis elements forever. We discuss the choice of basis in response to one of the questions of reviewer 9sbq, which addresses a similar point. In the case of the neural net described by the reviewer, if the agent can be understood as switching to a new basis element in light of experience, then the neural net is viewed as learning (that is, an agent updating its weights its not, own its own, sufficient to produce a learning agent—it must be that the agent actually updates its behavior in response to experience in the relevant way; and this could happen due to the recurrent state update, or due to weight change).
>
> > Finally, the paper would benefit from a more in-depth discussion of how previous approaches to continual RL compare to the proposed framework or can be embedded into it. Similarly, the authors mention several avenues for future research such as the examination of catastrophic forgetting - it might be interesting to include at least a characterisation of these problems in the new framework to inspire future work.
>
> These are both very good suggestions, thanks! We will add two pieces of discussion to the paper. First, as mentioned in the reply to all reviewers ("[Overall Response]"), we will add an in-depth discussion about the relationship between our definition of CRL and prior approaches to thinking about CRL, with an emphasis on the work by Ring and the recent CRL survey by Khetarpal et al. Succinctly, the work by Ring emphasizes the generality of the environment model and the reward function, while the survey by Khetarpal et al. focuses explicitly on non-stationarity, similar to work on multi-task RL. Our definition will directly accommodate and extend both of these views, and we are happy to include this discussion in the paper. Second, we can expand on connections to central empirical phenomena like catastrophic forgetting and in-context learning. For example, in-context learning might be fully characterized in terms of specific kinds of learning rules and agent bases (our definitions 3.1 and 3.2), where the base elements themselves are capable of more sophisticated learning. Plasticity loss can be captured in terms of changes to the capacity of an agent over time, as discussed in the general response— we believe these tools may help open new lines of study surrounding plasticity and related concepts. While these lines of research are still developing, we believe there is new opportunity raised to model and analyze these important empirical phenomena using the formal language from our work.

---

> > ### Comment · Reviewer_z8Jz · 2023-08-21
> >
> > I'd like to thank the authors for their thorough responses.
> > I think the proposed changes sufficiently alleviate most of the weaknesses and am updating my score accordingly.

---

### Author Rebuttal · Authors · 2023-08-08

**[Overall Response]**

First we would like to thank all of the reviewers for their time and energy in reading and commenting on our paper. We recognize this takes considerable effort, and we appreciate it

**Summary:** Overall, our impression of the reviews is that there is a lot of enthusiasm around the ambition, importance, novelty, and high writing quality of the work, as well as its potential to open new research pathways and perspectives. Many of the reviewers raise excellent questions and suggestions, and we believe they can each be addressed, and that doing so will strengthen the paper. We here briefly summarize the main points that were raised across several reviews, and provide a more detailed response to each individual reviewer below.

**Point 1: The abstract is too short.**

We had originally thought a simple abstract would be a good choice, but we hear the reviewers’ suggestion, and will modify our abstract accordingly. We will plan to change our abstract to the following:

_[Abstract draft]_

> In the standard view of the reinforcement learning problem, an agent interacts with an environment with the goal of efficiently identifying an optimal behavior. However, this perspective is based on a restricted view of _learning as finding a solution_, rather than treating learning as _endless adaptation_. Instead, _continual_ reinforcement learning refers to the setting in which the best agents keep learning forever. Despite the importance of this setting, the community lacks a simple, canonical definition of the problem that makes its primary commitments and concepts both precise and clear. To this end, this paper is dedicated to carefully defining the continual reinforcement learning problem. We formalize the notion of agents that “keep learning forever” through a pair of operators on agents that we call the “generates” and “reaches” operators that provide a new mathematical language for analyzing and cataloging agents. Using this new language, we define a continual learning agent as one that can be understood as carrying out an implicit search process indefinitely, and continual reinforcement learning as any setting in which the best agents are all continual learning agents. We provide two motivating examples of the setting, illustrating that traditional views of continual learning such as multi-task reinforcement learning and continual supervised learning are special cases of our definition. Furthermore, we establish necessary properties of both the continual reinforcement learning problem and the new operators. Collectively, these definitions, insights, and results formalize many intuitive concepts at the heart of continual learning, and open new research pathways surrounding continual learning agents.

We are happy to make changes to the above draft, and we hope this resolves the reviewers’ concern.

**Point 2: We could make additional contact with existing work on continual RL**

This is a very good suggestion. We will plan to add an expanded discussion in the paper commenting in-depth about the relationship between our definition of CRL and prior work on CRL, with an emphasis on the line of work by Ring and the recent CRL survey by Khetarpal et al. Succinctly, the works by Ring emphasizes the generality of the environment model and the reward function, while the survey by Khetarpal et al. focuses explicitly on non-stationarity, similar to work on multi-task RL by Wilson et al. (2007) and Brunskill and Li (2014). Our definition directly accommodates and extends both of these views, and we are happy to include a detailed discussion about this in the paper.

**Point 3: “Similarly, the authors mention several avenues for future research such as the examination of catastrophic forgetting - it might be interesting to include at least a characterisation of these problems in the new framework to inspire future work.”**

This is another great suggestion. While these lines of research are still in their early stages (we are actively exploring them now), we do believe there is a new opportunity to model and analyze these important empirical phenomena using the language from our work, and we will plan to expand on this discussion in the paper.

As one example of a connection to practical considerations, let’s consider plasticity loss. Plasticity loss can be well thought of using the conceptual tools we develop: _plasticity_ is the fraction of an agent’s corresponding agent basis that remains reachable to the agent over time, and plasticity _loss_ is any reduction of this number. We foresee opportunities to both diagnose agents from a new perspective, and to design algorithms based around this measure; for example, by developing learning rules (Defn. 3.2) that provably maintain plasticity (or analysing which existing learning rules maintain plasticity, and under what conditions).

Regarding pathways toward defining new algorithms, we have two responses.

First, as we argue after giving the definition of CRL (Defn. 4.1), our goal in defining CRL precisely is to encourage a departure in how we _think_ about designing agents: Given a basis, rather than try to build agents that can _solve_ problems by identifying a fixed high-quality element of the basis, we should instead focus on designing agents that continue to update their basis element indefinitely in light of experience. We believe this change in _perspective_ alone is needed, and can shape our basic approach to designing learning algorithms.

Second, as a slightly more concrete proposal, we might explicitly characterize learning rules that are guaranteed to produce a continual learning agent of the kind defined by Definition 4.1—what are the necessary and sufficient conditions of such a learning rule? Answering this question can give rise to algorithmic primitives that are guaranteed to produce continual learning agents (which, as Dohare et al. 2022 showed recently, is not true of many standard learning rules).

---

### Comment · Area_Chair_uQoa · 2023-08-19

To Authors:

Thank you for submitting your rebuttal. We appreciate your efforts in addressing the reviewers' comments and providing clarification.

To Reviewers:

We kindly request you to complete your response to the authors' rebuttal as soon as possible. Time is of the essence, and the deadline is fast approaching. Your timely feedback and expertise are crucial in ensuring a fair and thorough evaluation process. Please prioritize reviewing the authors' response and provide your final feedback accordingly.

---

### Decision · Program_Chairs · 2023-09-21

**Decision:**

Accept (poster)

**Comment:**

The reviewers mostly agreed that this is a solid paper with significant contributions.